# Outlier-Robust Phase Retrieval in Nearly-Linear Time

## Abstract

Phase retrieval is a fundamental problem in signal processing, where the goal is to recover a (complex-valued) signal from phaseless intensity measurements. In this paper, we propose and study the (real-valued) outlier-robust phase retrieval problem. Specifically, we seek to recover a vector $x \in \mathbb{R}^d$ from $n$ intensity measurements $y_i = (a_i^\top x)^2$, where a small fraction of the $(a_i, y_i)$ pairs are adversarially corrupted. Our main result is a near-sample-optimal and nearly-linear-time algorithm that provably recovers the ground-truth vector. Our algorithm first solves a lightweight convex program to find an initial point close to the ground truth, and then runs a robust version of gradient descent to achieve exact recovery. Our approach is conceptually simple and provides a framework for developing robust algorithms for other non-convex optimization problems.

## 1 Introduction

Phase retrieval is a fundamental problem in signal processing with applications in various fields, including electron microscopy [35], crystallography [36, 39], astronomy [13], and optical imaging [40]. In these applications, one often has access to only the magnitudes of the Fourier transforms of a complex signal. This is because measuring magnitude (e.g., by aggregating energy over time) is much easier than measuring phase (which requires detecting rapid changes). We refer the reader to the survey articles [40, 29] for more details about the theory and applications of phase retrieval.

In this paper, we focus on the (real-valued) generalized phase retrieval problem, where we are given intensity measurements of an arbitrary linear operator.

**Definition 1.1** (Phase Retrieval). *Let $x \in \mathbb{R}^d$ be the ground-truth vector. Let $a_1, \ldots, a_n \in \mathbb{R}^d$ be $n$ sampling vectors and let $y_i = \langle a_i, x \rangle^2 \in \mathbb{R}$ be the corresponding intensity measurements. Given $(a_i, y_i)_{i=1}^n$ as input, the task is to recover $x$.*

Note that it is impossible to distinguish between $x$ and $-x$, thus recovering either is sufficient. This problem has been extensively studied by the machine learning community. Under certain assumptions on the distribution of the sampling vectors, such as when the $a_i$'s are independent and Gaussian distributed, $\Theta(d)$ input pairs $(a_i, y_i)$ are necessary and sufficient for exact recovery [8]. Additionally, it has been shown that the problem can be solved in linear time with respect to the size of the input [8]. This was first achieved using semidefinite programming (SDP) relaxations [6]. In practice, the problem is often solved by applying first-order optimization methods (e.g., gradient descent) to a suitable objective function such as

$$\min_{z \in \mathbb{R}^d} \quad f(z) = \sum_{i=1}^n (y_i - \langle a_i, z \rangle^2)^2 . \tag{1}$$

Even though many natural formulations of the phase retrieval objective are nonconvex, including the one in (1), prior work has shown that, depending on the input distribution, they may not have spurious local optima and thus can be solved using first-order optimization methods [37, 4, 43].

However, these analyses of the objective landscape rely on strong assumptions, such as the sampling vectors $a_i$ being i.i.d. Gaussian. Our work is motivated by the following questions: Can we relax the assumptions used to prove landscape results for tractable nonconvex problems? In the context of phase retrieval, can we recover the ground-truth vector $x$ when a small subset of the $(a_i, y_i)$ pairs are adversarially corrupted?

Our focus on this adversarial setting, where an $\epsilon$-fraction of the input data is corrupted, is inspired by recent advances in high-dimensional robust statistics. An example problem in robust statistics is to estimate the mean of a $d$ dimensional spherical Gaussian when $\epsilon$-fraction of the samples are arbitrarily corrupted. The goal of high-dimensional statistics is often to design efficient algorithms that can achieve dimension-independent error guarantees. Early work in robust statistics [45, 26, 28] provided sample-efficient estimators for various tasks, but with runtimes exponential in the dimension. A recent line of work initiated by [16] and [31] has developed computationally efficient robust algorithms for many fundamental statistical and learning tasks. Significant progress has been made in the algorithmic aspects of robust high-dimensional statistics (see, e.g., [15]).

We now formally define the $\epsilon$-corruption model we study in this paper. For clarity, we define it directly in the context of phase retrieval.

**Definition 1.2** ($\epsilon$-Corrupted Set of Samples). *Let $x \in \mathbb{R}^d$ be the ground-truth vector. Let $\epsilon > 0$. First, the algorithm specifies the sample complexity $n$. Then, $n$ sampling vectors $(a_1, \ldots, a_n)$ are drawn from some known distribution $D$ and the corresponding intensity measurements $y_i = \langle a_i, x \rangle^2$ are calculated. The adversary is allowed to replace $\epsilon n$ pairs $(a_i, y_i)$ with arbitrary data. We call a set of $(a_i, y_i)$ pairs $\epsilon$-corrupted if it is generated by this process.*

In this paper, we say *samples* to refer to $(a_i, y_i)$ pairs. Note that we allow corruption in both the sampling vectors $a_i \in \mathbb{R}^d$ and the intensity measurements $y_i \in \mathbb{R}$, as long as the fraction of corruption is at most $\epsilon$. We now formally define outlier-robust phase retrieval, the main problem we study in this paper.

**Problem 1.3** (Outlier-Robust Phase Retrieval). *Let $x \in \mathbb{R}^d$ be the ground-truth with $\|x\|_2 = 1$. Let $\epsilon > 0$. Let $(a_1, \ldots, a_n)$ be $n$ sampling vectors drawn i.i.d. from $\mathcal{N}(0, I) \in \mathbb{R}^d$, and let $y_i = \langle a_i, x \rangle^2$ be the corresponding intensity measurements. An adversary arbitrarily corrupts an $\epsilon$-fraction of these $(a_i, y_i)$ pairs, and gives the $\epsilon$-corrupted set of samples as input to the algorithm. The task is to find a vector $z \in \mathbb{R}^d$ such that $\min\{\|z - x\|_2, \|z + x\|_2\} \leq \Delta$ for some precision parameter $\Delta > 0$.*

A simpler adversary setting where corruption is restricted to the intensity measurements $y_i$ has been studied previously [27, 51, 19]. In our work, we study a more comprehensive setting where we also allow corruption in the measurement vectors. This models realistic scenarios where the measurement process is affected by hardware noise, miscalibration, or adversarial tampering, leading to perturbations in the sampling vectors $a_i$ as well as in $y_i$. Concurrent work [3] studies this same problem. However, their algorithm uses a black-box subroutine for robust covariance estimation and thus requires at least $\Omega(d^2)$ time, making it impractical in high dimensions.

We are interested in designing a scalable and provably robust algorithm for Problem 1.3. We would like to resolve the following question:

> *Can we design a provably robust algorithm for the outlier-robust phase retrieval problem (Problem 1.3) that has near-optimal sample complexity and runs in nearly-linear time?*

## 1.1 Our Results

We answer the question outlined in the previous subsection affirmatively.

**Theorem 1.4** (Main). *Consider the outlier-robust phase retrieval problem as defined in Problem 1.3, where $x \in \mathbb{R}^d$ is the ground-truth vector. Let $0 < \epsilon \leq \epsilon_0$ for sufficiently small universal constant $\epsilon_0$. Let $\Delta > 0$ be the desired precision. Given an $\epsilon$-corrupted set of $n = \widetilde{\Omega}(d \log^2(1/\Delta))$ samples, we can compute $z \in \mathbb{R}^d$ in time $\widetilde{O}(nd)$ such that $\min(\|z - x\|_2, \|z + x\|_2) \leq \Delta$ with probability at least $0.95$.* [1]

---

[1] Throughout the paper, we use $\widetilde{O}(\cdot)$ and $\widetilde{\Omega}(\cdot)$ to hide logarithmic factors in their parameters.

84 A few remarks are in order regarding Theorem 1.4. First, even without corruption, phase retrieval
85 requires $\Omega(d)$ samples [8]. When $\Delta \geq \frac{1}{\text{poly}(d)}$, Theorem 1.4 requires $\widetilde{O}(d)$ samples, runs in nearly
86 linear time (since the input size is $O(nd)$), and achieves exact recovery. Therefore, our algorithm
87 simultaneously achieves the best possible error, sample complexity, and runtime (up to logarithmic
88 factors).

89 Second, the success probability in Theorem 1.4 can be boosted to $1 - \tau$ for any $\tau > 0$ by incurring
90 an additional $O(\log(1/\tau))$ factor in the sample complexity and runtime. This can be achieved, for
91 example, by partitioning the input, repeating the algorithm, letting candidate solutions vote for those
92 within distance $2\Delta$, and finally selecting the solution with the most votes.

93 Lastly, $\epsilon_0$ in Theorem 1.4 is an absolute constant that is independent of $n$ or $d$, and our algorithm
94 works for any corruption level $0 \leq \epsilon \leq \epsilon_0$. An important observation is that the optimal sample
95 complexity is $\Theta(d)$, which is independent of $\epsilon$. This follows from the fact that exact recovery is
96 possible as long as the clean samples provide enough constraints to fully determine $x$, which has
97 $d - 1$ degrees of freedom. [2] This is in contrast to problems in robust high-dimensional statistics, such
98 as robust mean estimation, where exact recovery is impossible with a finite number of samples (even
99 without corruption).

100 Since our algorithm guarantees exact recovery (to arbitrary precision $\Delta$) for any corruption level $\epsilon$ as
101 long as $\epsilon < \epsilon_0$, any input with corruption level $\epsilon < \epsilon_0$ can be treated as if it were corrupted at a fixed
102 level $\epsilon_0$. This explains why neither the sample complexity nor the runtime of our algorithm depends
103 on $\epsilon$. For simplicity, we refer to the corruption level $\epsilon$ as a sufficiently small universal constant
104 throughout the remainder of the paper unless otherwise noted.

## 1.2 Our Approach and Techniques

106 When there are infinitely many samples and no corruption, the objective function $f(z)$ simplifies to

$$f(z) = \mathop{\mathbb{E}}_{a \sim \mathcal{N}(0, I_d)} \left[ (\langle a_i, z \rangle^2 - y_i)^2 \right] = 3 \|x\|_2^4 + 3 \|z\|_2^4 - 2 \|x\|_2^2 \|z\|_2^2 - 4 \langle x, z \rangle^2 .$$

107 Despite being nonconvex, it is known that $f$ has no spurious local optima [37, 4, 43].

108 Our approach follows a two-step structure used in many local convergence results for nonconvex
109 problems (e.g., Candès et al. [4]), where the goal is to first initialize into a region free of saddle points
110 and then perform gradient descent. Both steps are vulnerable to adversarial attacks and we develop
111 provably robust and nearly-linear time algorithms for both steps. In the first step, we use spectral
112 techniques to obtain an initial guess that is sufficiently close to the ground truth. In the second step,
113 we run a robust gradient descent algorithm to refine this guess and converge to the final solution.

114 **Step 1: Robust Spectral Initialization.** Consider the following matrix $Y = \frac{1}{n} \sum_{i=1}^n y_i a_i a_i^\top$
115 where $y_i = \langle a_i, x \rangle^2$. When there is no corruption and the $a_i$'s are drawn i.i.d. from $\mathcal{N}(0, I)$,
116 we have $\mathbb{E}[Y] = I + 2xx^\top$. Hence, when there are enough samples and no corruption, we can
117 obtain a good estimate of the ground truth $x$ (or $-x$) by computing the largest eigenvector of $Y$.
118 However, the corrupted $(a_i, y_i)$ pairs can arbitrarily change the largest eigenvector of $Y$. One natural
119 approach, which was explored in concurrent work [3], is to apply known robust covariance estimation
120 algorithms [11, 1] to estimate $Y$. While this can recover the top eigenvector, the runtime is at least
121 $\Omega(d^2)$, which is too slow for our goal of designing a nearly linear time algorithm.

122 One of the main technical insights of our work is that it is not necessary to robustly estimate the entire
123 matrix $Y$, we only need to recover its few largest eigenspaces. We will assign a weight $w_i$ to each
124 sample such that the weighted intensity-based matrix $Y_w = \sum_{i=1}^n w_i y_i a_i a_i^\top$ is close to $I + 2xx^\top$.
125 We show that this can be done in nearly-linear time, which is highly non-trivial: the ground-truth
126 vector $x$ is unknown, and explicitly computing $Y_w$ via fast matrix multiplication takes $\Omega(d^2)$ time.

127 A key observation is that the corrupted samples can only add directions to $Y_w$ but cannot remove
128 directions, because each individual term $y_i a_i a_i^\top$ is a PSD matrix. (We assume w.l.o.g. that $y_i \geq 0$ for

---

[2]To build intuition why exact recovery is possible, consider the univariate case with an arbitrary positive
ground-truth $x \in \mathbb{R}$. In this case, one can compute the multiset $\{\sqrt{y_i/a_i^2}\}$, and as long as $\epsilon < 1/2$, its most
frequent element will be the correct $x$ with probability 1.

all $i$, because any input with $y_i < 0$ must be corrupted.) Consequently, if we can compute a weight vector $w$ that minimizes the sum of the top two eigenvalues of $Y_w$ (which is a convex optimization problem), we can recover a matrix that is close to the unknown unbiased expectation $I + 2xx^\top$. We show that this optimization problem can be solved in $\widetilde{O}(nd)$ time by leveraging algorithmic techniques developed for list-decodable mean estimation [12].

**Step 2: Robust Gradient Descent.** Starting with the initial guess $z$ given by the robust spectral initialization, we want to run gradient descent to recover the ground truth $x$. Without corruption, if the initialization is close enough to $x$, each iteration will bring $z$ closer to $x$ by a constant factor. Intuitively, approximating the gradient at a specific point amounts to a robust mean estimation problem (for the underlying distribution of the gradients). When the input data is $\epsilon$-corrupted, the gradients of the $n$ samples can be viewed as an $\epsilon$-corrupted set of vectors.

We can approximate the true gradient by running robust mean estimation on this $\epsilon$-corrupted set of $n$ gradients. To show convergence, the estimation of the gradient needs to be more accurate as we get closer to the optimal solution, and we show that this is possible because the variance of the gradient on clean samples decreases as the solution gets closer to the optimal solution.

### 1.3 Related and Prior Works

**Phase Retrieval.** Phase retrieval arises in various fields of science and engineering [13, 36]. Early research introduced error-reduction algorithms [25, 20, 21]. Convex and nonconvex optimization with various objective functions were later proposed and achieved exact recovery [47, 4–6, 48, 49, 41].

**Outlier-Robust Phase Retrieval.** Robust phase retrieval has been explored in the literature [51, 30, 8, 7, 34]. A simpler setting where corruption is restricted to the intensity measurements $y_i$ has been studied previously [27, 51, 19]. In Appendix C, we show that methods developed for this setting do not work in ours. Concurrent work [3] is the only one that studies the general corruption model that we consider, allowing corruption in both the sampling vectors $a_i$ and the intensity measurements $y_i$. The algorithm in [3] achieves near-optimal sample complexity, but relies on robust covariance estimation in a black-box manner, resulting in a slower runtime compared to ours. We emphasize that algorithm runs in nearly-linear time and achieves near-optimal sample complexity, which demonstrates that allowing corruption in both $a_i$ and $y_i$ incurs almost no penalty (asymptotically) in terms of statistical or computational complexity.

**Nonconvex Optimization.** Besides phase retrieval, it is known that all local optima are globally optimal for natural nonconvex formulations of various learning problems, such as matrix completion [24], matrix sensing [2], dictionary learning [42], and tensor decomposition [23] (we refer the interested reader to Chapter 7 of the book by [50]). A recent line of work explored the robustness of such landscape results: [33] studied matrix sensing in the $\epsilon$-corruption model, [9] and [22] studied semi-random matrix completion and matrix sensing.

**High-Dimensional Robust Statistics.** Recent works developed nearly-linear time algorithms for robust mean estimation [10, 18, 32]. The robust gradient descent algorithm we use is closely related to algorithms proposed in previous works for finding *first-order* stationary points in robust stochastic optimization [38, 17].

## 2 Preliminary and Background

**Notation.** Let $[n] = \{1, 2, \ldots, n\}$. For a vector $x$, we denote its $i^{th}$ coordinate by $x_i$. We use $\|x\|_1$, $\|x\|_2$, and $\|x\|_\infty$ to denote the $\ell_1$, $\ell_2$, and $\ell_\infty$ norm of $x$, respectively. For two vectors $x$ and $y$, we use $\langle x, y \rangle = x^\top y$ to denote their inner product.

We write $I$ for the identity matrix. For a matrix $A$, we use $\|A\|_2$ to denote its spectral norm. A symmetric matrix $A$ is positive semidefinite (PSD) if $x^\top A x \geq 0$ for all vectors $x$. For two symmetric matrices $A$ and $B$, we write $A \preceq B$ if $B - A$ is PSD. We write $\lambda_k(A)$ as the $k^{th}$ largest eigenvalue of $A$, and $\overline{\lambda}_k(A)$ as the sum of the $k$ largest eigenvalues of $A$. The Ky Fan $k$-norm of a matrix $A$ is the sum of its $k$ largest singular values, which is equal to $\overline{\lambda}_k(A)$ when $A$ is PSD.

**Ky Fan Norm Packing SDP.** In our robust spectral initialization step, we solve a Ky Fan norm packing semidefinite program (SDP) of the following form:

$$\max_{w \in \mathbb{R}^n_{\geq 0}} \quad \|w\|_1 \quad \text{subject to} \quad \sum_{i=1}^n w_i A_i \preceq I, \quad \overline{\lambda}_k \left( \sum_{i=1}^n w_i B_i \right) \leq k \qquad (**)$$

We use the nearly-linear time Ky Fan norm SDP solver from [12].

**Lemma 2.1** (Ky Fan Norm SDP Solver [12]). *Given an SDP (**) with positive semidefinite matrices $A_i \in \mathbb{R}^{d_1 \times d_1}$ and $B_i \in \mathbb{R}^{d_2 \times d_2}$ with $A_i = C_i C_i^\top$ and $B_i = D_i D_i^\top$ for all $i \in [n]$, integer $k > 0$, error tolerance $\epsilon_1 \geq 1/n^2$, and failure probability $\tau > 0$, one can in time $\widetilde{O}((t_C + t_D + d_1 + d_2) \operatorname{poly}(1/\epsilon_1, \log(1/\tau)))$ output $w' \in \mathbb{R}^n_{\geq 0}$ such that $\|w'\|_1 \geq (1 - \epsilon_1)\mathsf{OPT}$ with probability at least $1 - \tau$. Here $\mathsf{OPT}$ is the optimal value of (**), $t_{C_i}$ and $t_{D_i}$ are the time taken to perform a matrix-vector product with $C_i$ and $D_i$ respectively, and $t_C = \sum_{i=1}^n t_{C_i}$ and $t_D = \sum_{i=1}^n t_{D_i}$.*

**Top Eigenvector Computation.** We use the power method to compute an approximate top eigenvector. We refer to the analysis of the power method for PSD matrices by Trevisan [44].

**Lemma 2.2** (Top Eigenvector via Power Method [44]). *Let $A \in \mathbb{R}^{d \times d}$ be a PSD matrix. Let $\lambda_1$ be the largest eigenvalue of $A$. For any $\epsilon_2 > 0$, one can compute a unit vector $x \in \mathbb{R}^d$ in time $O((t_A + d) \log(d)/\epsilon_2)$ such that $x^\top A x \geq (1 - \epsilon_2)\lambda_1$ with probability at least $0.99$, where $t_A$ is the time taken to perform a matrix-vector multiplication with $A$.*

**Robust Mean Estimation.** In the robust gradient descent step, we use nearly-linear time robust mean estimation algorithms for bounded-covariance distributions [10, 14, 18] to approximate the true gradient.

**Lemma 2.3** (Robust Mean Estimation [18]). *Let $D$ be a distribution on $\mathbb{R}^d$ with unknown mean $\mu$ and unknown covariance matrix $\Sigma$ where $\Sigma \preceq \sigma^2 I$. Let $\epsilon_3 > 0$ be a sufficiently small universal constant. Let $0 < \epsilon \leq \epsilon_3$ and $\tau > 0$. Given an $\epsilon$-corrupted set of $n$ samples drawn from $D$, one can output a vector $\widehat{\mu} \in \mathbb{R}^d$ in time $\widetilde{O}(nd \log(1/\tau))$ such that, with probability at least $1 - \tau - \exp(-n\epsilon)$, we have $\|\widehat{\mu} - \mu\|_2 = O\left( \sqrt{\epsilon} + \sqrt{\frac{d}{n\tau}} + \sqrt{\frac{d(\log d + \log(1/\tau))}{n}} \right) \sigma$.*

# 3 Outlier-Robust Phase Retrieval

In this section, we present key technical lemmas for the two stages of our algorithm: robust spectral initialization (Lemma 3.1) and robust gradient descent (Lemma 3.2). We then use these lemmas to prove our main result (Theorem 1.4).

Lemma 3.1 shows that we can compute an initial guess close to the ground truth.

**Lemma 3.1** (Robust Spectral Initialization). *Consider the setting of Problem 1.3, where $x \in \mathbb{R}^d$ is the ground-truth. Let $\epsilon$ be a sufficiently small universal constant. Given an $\epsilon$-corrupted set of $n = \widetilde{\Omega}(d)$ samples, we can compute $z_1 \in \mathbb{R}^d$ in time $\widetilde{O}(nd)$ such that $\min\{\|z_1 - x\|_2, \|z_1 + x\|_2\} \leq \frac{1}{8}$ with probability at least $0.97$.*

Lemma 3.2 shows that after the initialization, a robust gradient descent algorithm can recover the ground-truth vector to arbitrary precision.

**Lemma 3.2** (Robust Gradient Descent). *Consider the setting of Problem 1.3, where $x \in \mathbb{R}^d$ is the ground-truth vector. Let $\Delta > 0$ be the desired precision. Let $\epsilon$ be a sufficiently small universal constant. Given an $\epsilon$-corrupted set of $n = \widetilde{\Omega}(d \log^2(1/\Delta))$ samples and an initial guess $z_1$ with $\|z_1 - x\|_2 \leq \frac{1}{8}$, we can compute a vector $z \in \mathbb{R}^d$ in time $\widetilde{O}(nd)$ such that $\|z - x\|_2 \leq \Delta$ with probability at least $0.98$.*

Lemma 3.1 is proved in Section 4. Lemma 3.2 is proved in Section 5. We first use these lemmas to prove Theorem 1.4.

*Theorem 1.4.* Let $\epsilon_0$ be the minimum of the two universal constants of Lemma 3.1 and Lemma 3.2. Let $0 \leq \epsilon \leq \epsilon_0$. We assume that we have access to two separate $\epsilon$-corrupted sets of samples, and use

one set for Lemma 3.1 and the other for Lemma 3.2. Formally, one could randomly partition the input samples into two sets and apply Chernoff bounds to show that both sets are $(2\epsilon)$-corrupted with high probability.

By Lemma 3.1, we can compute a vector $z_1 \in \mathbb{R}^d$ such that $\min\{\|z_1 - x\|_2, \|z_1 + x\|_2\} \leq \frac{1}{8}$ for the ground-truth vector $x$. Given $z_1$, by Lemma 3.2, we can output a vector $z \in \mathbb{R}^d$ such that $\min\{\|z - x\|_2, \|z + x\|_2\} \leq \Delta$ for the desired precision parameter $\Delta > 0$.

Let $n_1 = \widetilde{\Omega}(d)$ and $n_2 = \widetilde{\Omega}(d \log^2(1/\Delta))$ denote the number of samples used in Lemma 3.1 and Lemma 3.2, respectively. The overall sample complexity is therefore $n = n_1 + n_2 = \widetilde{\Omega}(d \log^2(1/\Delta))$. The overall runtime is $\widetilde{O}(n_1 d) + \widetilde{O}(n_2 d) = \widetilde{O}(nd)$. The overall success probability is at least $0.95$ by a union bound over Lemma 3.1 and Lemma 3.2. $\qquad\square$

# 4 Robust Spectral Initialization

In this section, we prove Lemma 3.1: given an $\epsilon$-corrupted set of samples $\{(a_i, y_i)\}_{i \in [n]}$, we can compute an initial guess $z_1 \in \mathbb{R}^d$ that is close to the ground truth $x$ or $-x$.

Consider the matrix $Y = \frac{1}{n} \sum_{i=1}^{n} y_i a_i a_i^\top$. When there is no corruption, where $a_i \sim \mathcal{N}(0, I)$ and $y_i = \langle a_i, x \rangle^2$, we have $\mathbb{E}[Y] = I + 2xx^\top$. However, the corrupted $(a_i, y_i)$'s can change $Y$ arbitrarily. To address this, we propose a nearly-linear time initialization step (Algorithm 1) that computes a nonnegative weight vector $w \in \mathbb{R}^n$ such that the weighted sum $Y_w = \sum_{i=1}^{n} w_i y_i a_i a_i^\top$ is close to $I + 2xx^\top$. Consequently, we can show that the largest eigenvector of $Y_w$ is close to $\pm x$.

Let $G \subset [n]$ be the set of indices of the remaining good samples. An ideal approach would be to assign weight $\frac{1}{|G|} = \frac{1}{(1-\epsilon)n}$ to every sample in $G$, and weight $0$ to the corrupted samples. Formally, we consider weight vectors $w$ in the set $\Delta_{n,\epsilon} = \left\{ w \in \mathbb{R}_{\geq 0}^n : \|w\|_1 = 1 \text{ and } \|w\|_\infty \leq \frac{1}{(1-\epsilon)n} \right\}$. Algorithm 1 computes a near-optimal solution $\widehat{w}$ to the following optimization problem:

$$\min_{w \in \Delta_{n,\epsilon}} \lambda_1(Y_w) + \lambda_2(Y_w) ,$$

and then returns the largest eigenvector of $Y_{\widehat{w}}$.

---

**Algorithm 1:** Robust Spectral Initialization

**Input:** $\epsilon$-corrupted set of $n$ samples $\{(a_i, y_i)\}_{i \in [n]}$.
**Output:** An initial guess $z_1 \in \mathbb{R}^d$ of the ground-truth $x$ s.t. $\min\{\|z_1 - x\|_2, \|z_1 + x\|_2\} \leq \frac{1}{8}$.

1   $\widehat{w} \leftarrow$ a near-feasible, near-optimal solution to: $\min_{w \in \Delta_{n,\epsilon}} [\lambda_1(Y_w) + \lambda_2(Y_w)]$ using Lemma A.2, where $Y_w = \sum_{i=1}^{n} w_i y_i a_i a_i^\top$;
2   $z_1 \leftarrow$ an approximate top eigenvector of $Y_{\widehat{w}}$ using the power method (Lemma 2.2);

3   **return** $z_1$;

---

To prove that the largest eigenvector of $Y_{\widehat{w}}$ is close to $x$, we will show that $x^\top Y_{\widehat{w}} x$ is large, and there is a gap between the first and second largest eigenvalues of $Y_{\widehat{w}}$.

**Lemma 4.1.** *Consider the setting of Problem 1.3, where $x \in \mathbb{R}^d$ is the ground-truth vector. Fix $\delta > 0$. There exists constants $\epsilon(\delta)$ and $c(\delta)$ such that if we are given an $\epsilon(\delta)$-corrupted set of $n = \widetilde{\Omega}(c(\delta)d)$ samples $(a_i, y_i)_{i \in [n]}$, Algorithm 1 outputs $\widehat{w} \in \mathbb{R}^n$ in time $\widetilde{O}(nd \operatorname{poly}(1/\epsilon(\delta)))$ such that, with probability at least $0.98$, the following conditions hold:*

$$x^\top Y_{\widehat{w}} x \geq 3 - O(\delta), \quad |\lambda_1(Y_{\widehat{w}}) - 3| \leq O(\delta), \text{ and } \quad |\lambda_2(Y_{\widehat{w}}) - 1| \leq O(\delta).$$

*where $Y_{\widehat{w}} = \sum_{i=1}^{n} \widehat{w}_i y_i a_i a_i^\top$.*

We defer the proof of Lemma 4.1 to Appendix A.1 and first use it to prove the correctness and runtime of Algorithm 1 (Lemma 3.1).

*Proof of Lemma 3.1.* Let $Y_{\widehat{w}} = \sum_{i=1}^{d} \lambda_i v_i v_i^\top$ be the eigendecomposition of $Y_{\widehat{w}}$, where $\lambda_1 \geq \lambda_2 \geq \ldots \geq \lambda_d$. Let $x = \sum_{i=1}^{d} \alpha_i v_i$. Note that $\sum_{i=1}^{d} \alpha_i^2 = \|x\|_2^2 = 1$ and $|\alpha_i| \leq 1$. If the event of Lemma 4.1 is true, with $\epsilon < \epsilon(\delta)$ and $m = \widetilde{\Omega}(c(\delta)d)$, then

$$\begin{aligned} 3 - O(\delta) \leq x^\top Y_{\widehat{w}} x = \sum_{i=1}^{d} \lambda_i \alpha_i^2 &\leq \lambda_1 \alpha_1^2 + \lambda_2(1 - \alpha_1^2) \\ &\leq (3 + O(\delta))\alpha_1^2 + (1 + O(\delta))(1 - \alpha_1^2) \leq 1 + O(\delta) + 2\alpha_1^2 \ . \end{aligned}$$

This implies $|\alpha_1| \geq \alpha_1^2 \geq 1 - O(\delta)$. Consequently,

$$\begin{aligned} \min\{\|v_1 - x\|_2^2, \|v_1 + x\|_2^2\} &= \min\{(1 - \alpha_1)^2, (1 + \alpha_1)^2\} + \sum_{i=2}^{d} \alpha_i^2 \\ &= \min\{2 - 2\alpha_1, 2 + 2\alpha_1\} = 2 - 2|\alpha_1| \leq O(\delta) \ . \end{aligned}$$

We choose $\delta$ as a sufficiently small constant so that $\min\{\|v_1 - x\|_2, \|v_1 + x\|_2\} \leq O(\sqrt{\delta}) \leq \frac{1}{16}$. Note that since it is sufficient to choose $\delta$ as a small universal constant, $c(\delta)$ and $\epsilon(\delta)$ can also be treated as universal constants.

We use the power method to approximate the largest eigenvector $z_1$ of $Y_{\widehat{w}}$. By Lemma 2.2, $\|z_1\|_2 = 1$ and $z_1^\top Y_{\widehat{w}} z_1 \geq (1 - \epsilon_2)\lambda_1$. Choosing $\epsilon_2 = O(\delta)$ and using Lemma 4.1, we have $z_1^\top Y_{\widehat{w}} z_1 \geq 3 - O(\delta)$. By the same arguments, we can show that $\min\{\|v_1 - z_1\|_2, \|v_1 + z_1\|_2\} \leq \frac{1}{16}$, and then by the triangle inequality, we conclude that $\min\{\|z_1 - x\|_2, \|z_1 + x\|_2\} \leq \frac{1}{8}$.

The success probability of Algorithm 1 is at least 0.97, as $\widehat{w}$ satisfies Lemma 4.1 with probability at least 0.98, and the power method succeeds with probability at least 0.99 by Lemma 2.2.

The required number of samples is $n = \widetilde{\Omega}(d)$ by Lemma 4.1. Algorithm 1 runs in time $\widetilde{O}(nd)$: It takes time $\widetilde{O}(nd\,\text{poly}(1/\epsilon(\delta))) = \widetilde{O}(nd)$ to compute $\widehat{w}$ by Lemma 4.1. The power method can approximate the largest eigenvector of $Y_{\widehat{w}}$ in time $O(nd \log(d)/\delta) = \widetilde{O}(nd)$ by Lemma 2.2, since the matrix-vector product $Y_{\widehat{w}} v = \sum_{i=1}^{n} \widehat{w}_i y_i \langle a_i, v \rangle a_i$ can be computed in time $O(nd)$ for any $v \in \mathbb{R}^d$. $\qquad\square$

## 5 Robust Gradient Descent

After the robust initialization in Section 4, we have an initial guess $z_1 \in \mathbb{R}^d$ that is close to the ground truth $x$ or $-x$. We can assume without loss of generality that $z_1$ is closer to $x$ than to $-x$.

In this section, we prove Lemma 3.2: Given an initial guess $z_1$ with $\|z_1 - x\|_2 \leq \frac{1}{8}$, we can use a robust gradient descent algorithm (Algorithm 2) to recover $x$ to any desired precision $\Delta > 0$. We will show that Algorithm 2 converges geometrically even when the input is $\epsilon$-corrupted.

Consider the following nonconvex optimization problem:

$$\min_{z \in \mathbb{R}^d} \sum_{i=1}^{n} f_i(z) \quad \text{where} \quad f_i(z) = \left( \langle a_i, z \rangle^2 - y_i \right)^2 \ .$$

Let $g_i$ denote the gradient of $f_i$ with respect to $z$. Let $\mathcal{D}_z$ denote the distribution of $g_i(z)$ when there is no corruption. Formally, $g(z) \sim \mathcal{D}_z$ is distributed as

$$g(z) = \frac{\partial}{\partial z}\left[ \left( \langle a, z \rangle^2 - \langle a, x \rangle^2 \right)^2 \right] = 4\left( \langle a, z \rangle^2 - \langle a, x \rangle^2 \right) \langle a, z \rangle a \quad \text{where} \quad a \sim \mathcal{N}(0, I) \ . \quad (2)$$

To perform gradient descent, we want to approximate the *expected true gradient*

$$\mu_z = \mathop{\mathbb{E}}_{g(z) \sim \mathcal{D}_z} [g(z)] = \left( 12 \|z\|_2^2 - 4 \|x\|_2^2 \right) z - 8 \langle x, z \rangle x \ . \quad (3)$$

However, the input $\{(a_i, y_i)\}_{i \in [n]}$ is $\epsilon$-corrupted, so the corresponding gradients $\{g_i(z)\}_{i \in [n]}$ are an $\epsilon$-corrupted set of vectors drawn from $\mathcal{D}_z$. We will run nearly-linear time robust mean estimation algorithms (e.g., [18]) on $\{g_i(z)\}_{i \in [n]}$ to approximate the true gradient $\mu_z$.

The accuracy of robust mean estimation algorithms depends on the covariance matrix $\Sigma_z$ of the distribution $\mathcal{D}_z$. The following lemma upper bounds the spectral norm of $\Sigma_z$.

---

**Algorithm 2:** Robust Gradient Descent

---

**Input:** An $\epsilon$-corrupted set of $n$ samples $\{(a_i, y_i)\}_{i \in [n]}$, an initial guess $z_1$ with $\|z_1 - x\| \leq \frac{1}{8}$, and desired precision $\Delta > 0$.

**Output:** $z \in \mathbb{R}^d$ such that $\|z - x\|_2 \leq \Delta$, where $x$ is the ground-truth vector.

1   $T \leftarrow O(\log(1/\Delta))$, $\eta \leftarrow \frac{1}{300}$;

2   $\{N_1, \ldots, N_T\} \leftarrow$ a random disjoint partition of $[n]$ such that $|N_t| = \frac{n}{T}$ for all $t \in [T]$;

3   **for** $t = 1, 2, \ldots, T$ **do**

4      $g_i(z_t) \leftarrow 4 \left( \langle a_i, z_t \rangle^2 - y_i \right) \langle a_i, z_t \rangle a_i$;

5      $\widehat{\mu}_{z_t} \leftarrow$ robust mean estimation on input $\{g_i(z_t)\}_{i \in N_t}$ using Lemma 5.2;

6      $z_{t+1} \leftarrow z_t - \eta \widehat{\mu}_{z_t}$;

7   **end**

8   **return** $z_{T+1}$;

---

**Lemma 5.1.** *Let $x \in \mathbb{R}^d$ be the ground-truth vector. Let $\mathcal{D}_z$ be the distribution of gradients at $z$ as defined in Equation (2). For any $z$ with $\|z - x\|_2 \leq 1$, the covariance matrix $\Sigma_z$ of $\mathcal{D}_z$ satisfies*

$$\Sigma_z \preceq O\left( \|z - x\|_2^2 \right) I .$$

We defer the proof of Lemma 5.1 to Appendix B. For technical reasons, we randomly partition the input $\{(a_i, y_i)\}_{i \in [n]}$ into $T$ subsets and use one subset in each iteration. With high probability, each partition has at most $(2\epsilon)$-fraction of corrupted samples. The next lemma shows that, given the covariance bound in Lemma 5.1, we can approximate the true gradient $\mu_z$ from a $(2\epsilon)$-corrupted set of gradients with a small error.

**Lemma 5.2.** *Let $x \in \mathbb{R}^d$ be the ground-truth vector. Consider any $z \in \mathbb{R}^d$ with $\|z - x\|_2 \leq 1$. Let $\mathcal{D}_z$ be the distribution defined in Equation (2) and let $\mu_z$ be the mean of $\mathcal{D}_z$. Let $c > 0$, and let $\tau \in (0, 1/4)$ be a small constant. There exists a constant $\epsilon(c)$ such that given a $2\epsilon(c)$-corrupted set of $m = \Omega(d \log(d)/(c^2 \tau))$ vectors drawn from $\mathcal{D}_z$, we can compute $\widehat{\mu}_z \in \mathbb{R}^d$ in time $\widetilde{O}(md \log(1/\tau))$ such that $\|\widehat{\mu}_z - \mu_z\|_2 \leq c \|z - x\|_2$ with probability at least $1 - 2\tau$.*

*Proof of Lemma 5.2.* We constraint $\epsilon(c) \leq \epsilon_3/2$, where $\epsilon_3$ is the universal constant in the robust mean estimation algorithm stated in Lemma 2.3. By Lemma 5.1, the covariance matrix $\Sigma_z$ of $\mathcal{D}_z$ satisfies $\Sigma_z \preceq O\left( \|z - x\|_2^2 \right) I$. Lemma 2.3 guarantees that robust mean estimation returns a vector $\widehat{\mu}_z \in \mathbb{R}^d$ such that

$$\|\widehat{\mu}_z - \mu_z\|_2 \leq O\left( \sqrt{2\epsilon(c)} + \sqrt{\tfrac{d}{m\tau}} + \sqrt{\tfrac{d(\log d + \log(1/\tau))}{m}} \right) \sqrt{\|\Sigma_z\|_2} \leq c \|z - x\|_2 .$$

The last inequality follows by properly choosing the constant $\epsilon(c) = O(c^2)$. By Lemma 2.3, the runtime is $\widetilde{O}(md \log(1/\tau))$ and the success probability is at least $1 - \tau - \exp(-\epsilon_0 m) \geq 1 - 2\tau$. $\quad\square$

The next lemma shows that the approximate gradient from Lemma 5.2 is sufficient for gradient descent to converge, reducing the distance to the ground truth $x$ by a constant factor in each iteration. We provide a proof sketch for Lemma 5.3 and defer the full proof to Appendix B.

**Lemma 5.3.** *Let $x \in \mathbb{R}^d$ be the ground-truth vector. Suppose at iteration $t$ of Algorithm 2, the current solution $z_t$ satisfies $\|z_t - x\|_2 \leq \frac{1}{8}$. Let $\mu_{z_t}$ denote the expected true gradient at $z_t$ defined in Equation (3). Suppose the estimated gradient $\widehat{\mu}_{z_t} \in \mathbb{R}^d$ satisfies $\|\widehat{\mu}_{z_t} - \mu_{z_t}\|_2 \leq c \|z_t - x\|_2$ for $c = 4$. Then, we have*

$$\|z_{t+1} - x\|_2^2 \leq 0.99 \|z_t - x\|_2^2 .$$

*Proof Sketch of Lemma 5.3.* Even though the objective function is nonconvex, it is known that gradient descent is well-behaved when initialized close enough to a global optimum [37]. More specifically,

for any $z$ with $\|z - x\|_2 \leq \frac{1}{8}$, we can show that the expected true gradient at $z$ aligns with the direction $(z - x)$:

$$\langle \mu_z, z - x \rangle \geq 7.5 \|z - x\|_2^2 \quad \text{and} \quad \|\mu_z\|_2 \leq 29 \|z - x\|_2 \ ,$$

which is sufficient for proving geometric convergence.

Note that this analysis is robust to small error in $\mu_z$. When $\|\widehat{\mu}_z - \mu_z\|_2 \leq c \|z - x\|_2$, we have

$$\langle \widehat{\mu}_z, z - x \rangle \geq (7.5 - c) \|z - x\|_2^2 \quad \text{and} \quad \|\widehat{\mu}_z\|_2 \leq (29 + c) \|z - x\|_2 \ .$$

When $c < 7.5$, we can choose an appropriate step size $\eta$ such that the distance between $z$ and $x$ decreases by a constant factor in each iteration. $\qquad\square$

We are now ready to prove Lemma 3.2, which states the correctness and runtime of Algorithm 2.

*Lemma 3.2.* First, we analyze the success probability of Algorithm 2. Algorithm 2 can fail in two ways: *(i)* some $N_t$ has more than $(2\epsilon)$-fraction of corrupted samples, or *(ii)* robust gradient estimation fails in some iteration $t$. The probability of event *(i)* is at most $0.01$ for our choice of $n$, which follows from a standard application of Hoeffding's inequality and a union bound. For event *(ii)*, we choose $\tau \leq 0.005/T$ in Lemma 5.2, so each robust gradient estimation fails with probability at most $2\tau = 0.01/T$. By a union bound over $T$ iterations, the probability of event *(ii)* is at most $0.01$. For the rest of the proof, we assume these bad events do not happen.

Next, we prove the correctness of Algorithm 2. Since $\|z_1 - x\|_2 \leq \frac{1}{8}$, we can use Lemma 5.2 to obtain an approximation $\widehat{\mu}_{z_1}$ of the true gradient $\mu_{z_1}$ such that $\|\widehat{\mu}_{z_1} - \mu_{z_1}\|_2 \leq c \|z_1 - x\|_2$ with $c = 4$. Then, by Lemma 5.3, we have $\|z_2 - x\|_2 \leq 0.99 \|z_1 - x\|_2$ after one iteration of gradient descent. Applying these two lemmas repeatedly, after $T = O(\log(1/\Delta))$ iterations, we have $\|z_{T+1} - x\|_2 \leq \Delta$.

Finally, we analyze the sample complexity and runtime of Algorithm 2. The algorithm requires in total $n = mT = \Omega(d \log d \log^2(1/\Delta))$ samples. A random partition can be computed in $O(n)$ time via random shuffling. In each iteration, the $m$ gradients in $N_t$ can be computed in time $O(md)$ using Equation (2). By Lemma 5.2, the true gradient can be robustly estimated in time $\widetilde{O}(md \log(1/\tau)) = \widetilde{O}(md \log T) = \widetilde{O}(md \log \log(1/\Delta))$, and $z_t$ can be updated in time $O(d)$. The overall runtime of Algorithm 2 is $\widetilde{O}(n + Tmd \log \log(1/\Delta)) = \widetilde{O}(nd)$. $\qquad\square$

**Remark.** There are two technical details worth noting. First, robust mean estimation algorithms (Lemma 2.3) require a *known* upper bound $\sigma^2$ on the spectral norm of the covariance matrix of $\mathcal{D}_z$. By Lemma 5.1, a *known* upper bound on $\|z - x\|_2$ suffices. We can indeed maintain such an upper bound, which starts at $\frac{1}{8}$ and decreases geometrically, as shown in Lemma 5.3. Second, at each iteration $t$, to apply Lemma 5.2 to robustly estimate the gradient at $z_t$, we need a $(2\epsilon)$-corrupted set of gradients drawn from $\mathcal{D}_{z_t}$. This is why we use a set of fresh samples $N_t$. By the principle of deferred decisions, we can view $(a_i, y_i)_{i \in N_t}$ as being generated and corrupted *after* $z_t$ is chosen.

# 6 Conclusions and Future Directions

In this paper, we propose and study the problem of outlier-robust phase retrieval, where a small fraction of the input data is corrupted. Importantly, we allow adversarial corruption in both the sampling vectors $a_i \in \mathbb{R}^d$ and the intensity measurements $y_i \in \mathbb{R}$. We present a near-sample-optimal and nearly-linear-time algorithm for this problem with provable guarantees. One conceptual contributions of our work is that phase retrieval can be solved using a robust first-order methods even when the input is slightly misspecified or corrupted. Our algorithmic framework provides a general approach for developing robust algorithms for a wide range of tractable nonconvex problems, by first robustly initializing into a region free of saddle points and then using robust gradient descent to converge to a global optimum.

An immediate technical question is whether our sample complexity can be tightened by removing the $\log(1/\Delta)$ factors. One potential approach is to examine the stability conditions required by robust mean estimation algorithms and see if these conditions can be proved without using fresh samples in each iteration.

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

# A  Omitted Proofs in Section 4

## A.1  Proof of Lemma 4.1

In this section, we prove Lemma 4.1: We can compute $\widehat{w} \in \mathbb{R}^n$ in nearly-linear time such that $x^\top Y_{\widehat{w}} x$ is large and the two largest eigenvalues of $Y_{\widehat{w}}$ are approximately 3 and 1.

The following lemma shows that, for any $w \in \Delta_{n,2\epsilon}$, the contribution of the remaining good samples to $Y_w$ is close to $I + 2xx^\top$. This allows us to lower bound $\lambda_1(Y_{\widehat{w}})$, $\lambda_2(Y_{\widehat{w}})$, and $x^\top Y_{\widehat{w}} x$.

**Lemma A.1.** *Let $x \in \mathbb{R}^d$ be the ground-truth vector. Fix $\delta > 0$. There exists constants $\overline{\epsilon}(\delta)$ and $c(\delta)$ such that if we let $0 < \epsilon \leq \overline{\epsilon}(\delta)$, and we are given an $\epsilon$-corrupted set of $n = \widetilde{\Omega}(c(\delta)d)$ samples $(a_i, y_i)_{i \in [n]}$, with probability at least 0.99, for all $w \in \Delta_{n,2\epsilon}$,*

$$\left\| Y_{G,w} - (I + 2xx^\top) \right\|_2 \leq \delta ,$$

*where $G$ is the set of indices of the remaining good samples and $Y_{G,w} = \sum_{i \in G} w_i y_i a_i a_i^\top$.*

Intuitively, Lemma A.1 holds because the moments of Gaussian distributions are stable when a small fraction of the samples are removed. We defer the proof of Lemma A.1 to Appendix A.

The next lemma shows that, assuming Lemma A.1 holds, we can compute a near-optimal $\widehat{w}$ that minimizes $\lambda_1(Y_{\widehat{w}}) + \lambda_2(Y_{\widehat{w}})$ in nearly-linear time.

**Lemma A.2.** *Let $\delta \in (0, 1/2)$. There exists a constant $\epsilon(\delta)$ such that if we are given an $\epsilon(\delta)$-corrupted set of $n$ samples $(a_i, y_i)_{i \in [n]}$ that satisfies Lemma A.1, we can compute $\widehat{w} \in \Delta_{n,2\epsilon(\delta)}$ in time $\widetilde{O}(nd \operatorname{poly}(1/\epsilon(\delta)))$ such that with probability at least 0.99,*

$$\lambda_1(Y_{\widehat{w}}) + \lambda_2(Y_{\widehat{w}}) \leq 4 + O(\delta) ,$$

*where $d$ is the dimension of $a_i$ and $Y_{\widehat{w}} = \sum_{i=1}^n \widehat{w}_i y_i a_i a_i^\top$.*

*Proof.* We reduce the optimization problem $\min_{w \in \Delta_{n,\epsilon(\delta)}} [\lambda_1(Y_{\widehat{w}}) + \lambda_2(Y_{\widehat{w}})]$ to the following Ky Fan norm packing semidefinite program (SDP):

$$\max_{w \in \mathbb{R}_{\geq 0}^n} \quad \|w\|_1 \quad \text{subject to} \quad \sum_{i=1}^n w_i A_i \preceq I, \quad \overline{\lambda}_2\left(\sum_{i=1}^n w_i B_i\right) \leq 4 + O(\delta) \qquad (*)$$

in which $A_i = (1 - \epsilon(\delta)) n e_i e_i^\top$ where $e_i \in \mathbb{R}^n$ is the $i^{th}$ basis vector, $B_i = y_i a_i a_i^\top$, and $\overline{\lambda}_2$ is the sum of the two largest eigenvalues.

Let $G$ be the set of indices of the remaining good samples. Consider the weight vector $w^\star \in \mathbb{R}^n$ that is uniform on $G$:

$$w_i^* = \begin{cases} \frac{1}{|G|} = \frac{1}{(1-\epsilon)n} & i \in G , \\ 0 & i \notin G . \end{cases}$$

Let $\epsilon(\delta) = \min\{\delta, \overline{\epsilon}(\delta)\}$, where $\overline{\epsilon}(\delta)$ is the constant of Lemma A.1. By Lemma A.1, $Y_{w^\star} \preceq (1 + \delta)I + 2xx^\top$, which implies $\overline{\lambda}_2(Y_{w^\star}) \leq 4 + O(\delta)$. Since $w^\star$ is feasible, the optimal value OPT of (*) must be at least $\|w^\star\|_1 = 1$.

We invoke Lemma 2.1 to solve (*) with failure probability $\tau = 0.01$ and error tolerance parameter $\epsilon_1 = \epsilon(\delta)$. The resulting solution $w' \in \mathbb{R}_{\geq 0}^n$ satisfies $\|w'\|_1 \geq (1 - \epsilon_1)\text{OPT} \geq 1 - \epsilon(\delta)$. Define $\widehat{w} = \frac{w'}{\|w'\|_1}$ so that $\|\widehat{w}\|_1 = 1$. The constraint with $A_i$ guarantees that $\|w'\|_\infty \leq \frac{1}{(1-\epsilon(\delta))n}$, so $\|\widehat{w}\|_\infty \leq \frac{\|w'\|_\infty}{1-\epsilon(\delta)} \leq \frac{1}{(1-2\epsilon(\delta))n}$ and thus $w \in \Delta_{n,2\epsilon(\delta)}$. The constraint with $B_i$ implies that $\overline{\lambda}_2(Y_{w'}) \leq 4 + O(\delta)$, and after scaling we have $\overline{\lambda}_2(Y_{\widehat{w}}) \leq \frac{4+O(\delta)}{1-\epsilon(\delta)} \leq 4 + O(\delta)$ since $\epsilon(\delta) \leq \delta < \frac{1}{2}$. The success probability is at least $1 - \tau = 0.99$.

We can write $A_i = C_i C_i^\top$ with $C_i = \sqrt{(1-\epsilon)n} e_i$, and $B_i = D_i D_i^\top$ with $D_i = \sqrt{y_i} a_i$. It takes $O(1)$ time to perform a matrix-vector product with $C_i$, and $O(d)$ time with $D_i$. Therefore, $t_C + t_D$ in Lemma 2.1 is $O(nd)$, and the runtime of Lemma 2.1 is $\widetilde{O}(nd \operatorname{poly}(1/\epsilon))$ for $\tau = 0.01$. $\qquad \square$

We now proceed to prove Lemma 4.1.

*Proof of Lemma 4.1.* Without loss of generality, we can assume that $y_i \geq 0$ for all $i \in [n]$. Because $y_i$ should be $\langle a_i, x \rangle^2$, any sample with $y_i < 0$ must be corrupted and can be discarded. By Lemma A.2, we can compute $\widehat{w} \in \Delta_{n,2\epsilon(\delta)}$ such that $\lambda_1(Y_{\widehat{w}}) + \lambda_2(Y_{\widehat{w}}) \leq 4 + O(\delta)$.

By Lemma A.1 and $y_i \geq 0$, we have

$$Y_{\widehat{w}} = \sum_{i \in S} \widehat{w}_i y_i a_i a_i^\top \succeq \sum_{i \in G} \widehat{w}_i y_i a_i a_i^\top = Y_{G,\widehat{w}} \succeq (1 - \delta)I + 2xx^\top \ ,$$

which gives a lower bound $x^\top Y_{\widehat{w}} x \geq 3 - O(\delta)$, as well as lower bounds on the eigenvalues of $Y_{\widehat{w}}$:

$$\lambda_1(Y_{\widehat{w}}) \geq 3 - O(\delta) \quad \text{and} \quad \lambda_2(Y_{\widehat{w}}) \geq 1 - O(\delta) \ .$$

Putting the upper and lower bounds together, we obtain

$$\lambda_1(Y_{\widehat{w}}) = \overline{\lambda}_2(Y_{\widehat{w}}) - \lambda_2(Y_{\widehat{w}}) \leq 4 + O(\delta) - (1 - O(\delta)) \leq 3 + O(\delta) \ , \text{ and}$$
$$\lambda_2(Y_{\widehat{w}}) = \overline{\lambda}_2(Y_{\widehat{w}}) - \lambda_1(Y_{\widehat{w}}) \leq 4 + O(\delta) - (3 - O(\delta)) \leq 1 + O(\delta) \ .$$

Lemma A.1 holds with probability at least 0.99. Assuming Lemma A.1 holds, Lemma 2.1 succeeds with probability at least 0.99, so the success probability of Lemma 4.1 is at least 0.98.

For the initialization step, it suffices to use Lemma A.2 to find a $(1 - \epsilon(\delta))$-optimal solution $\widehat{w} \in \Delta_{n,2\epsilon(\delta)}$ to the SDP (*), and the runtime to compute such $\widehat{w}$ is $\widetilde{O}(nd \operatorname{poly}(1/\epsilon(\delta)))$. $\qquad\square$

## A.2   Proof of Lemma A.1

This section is devoted to the proof of Lemma A.1. We will use the following concentration results.

**Lemma A.3** ([4], Section A.4.2)**.** *Let $x \in \mathbb{R}^d$. For any $\delta > 0$, there exists a constant $C(\delta) > 0$ such that when $n > C(\delta) \cdot d \log d$ and we are given a set of $n$ samples $\{(a_i, y_i)\}_{i=1}^n$ with $a_i \sim \mathcal{N}(0, I)$ independently and $y_i = \langle a_i, x \rangle^2$ for all $i \in [n]$, then with probability at least 0.99, it holds*

$$\left\| \frac{1}{n} \sum_{i=1}^n y_i a_i a_i^\top - (I + 2xx^\top) \right\|_2 \leq \delta \ .$$

**Proposition A.4.** *Let $\alpha \in (0, 2/e)$. Let $X_1, \ldots, X_m$ be $m$ random variables drawn i.i.d. from $\mathcal{N}(0, 1)$. Define*

$$H = \{i \in [m] : |X_i| \geq 4 \ln^2(2/\alpha)\} \ .$$

*With probability at least $1 - 10^{-3}$, the following are all true:*

$$(a) \quad |H| \leq O(m\alpha) \ ,$$
$$(b) \quad \sum_{i \in H} X_i^4 = O\big(m\alpha \log^8(2/\alpha)\big) \ ,$$
$$(c) \quad \max_{i \in H} X_i^2 = O(\log m) \ .$$

*Proof.* For $X \sim \mathcal{N}(0, 1)$ and $t > 0$, it holds that $\mathbf{Pr}[|X| \geq t] \leq 2 \exp(-t^2/2)$. By setting $t = 4 \ln^2(2/\alpha)$, we have $\mathbf{Pr}[X^4 \geq t] \leq \alpha$. Let $Y_i \in \{0, 1\}$ be the indicator random variable for the event "$i \in H$", so that $|H| = \sum_{i \in H} Y_i$.

Because $\mathbb{E}\big[\sum_{i \in H} Y_i\big] \leq m\alpha$, by Markov's inequality, we have $|H| = O(m\alpha)$ with probability at least $1 - 10^{-3}/3$. We assume this holds for the rest of the proof.

By the principle of deferred decisions, an equivalent way to draw $X_1, \ldots, X_m$ from $\mathcal{N}(0, 1)$ is to first draw $Y_i$, and then draw $X_i$ conditioned on the value of $Y_i$. Note that $H$ is fixed after drawing $Y_1, \ldots, Y_m$.

$$\mathbb{E}\left[\sum_{i \in H} X_i^4\right] = \sum_{i \in H} \mathbb{E}\big[X_i^4 \mid |X_i| \geq 4 \ln^2(2/\alpha)\big] \leq O(m\alpha) \, \mathbb{E}\big[X_i^4 \mid |X_i| \geq 4 \ln^2(2/\alpha)\big] \ . \quad (4)$$

For any $t \geq 4$, by the definition of conditional expectation, and using the fact that $X_i$ is normally distributed:

$$\mathbb{E}\big[X_i^4 \mid |X_i| \geq t\big] = \frac{\frac{2}{\sqrt{2\pi}} \int_t^\infty x^4 e^{-x^2/2} dx}{\frac{2}{\sqrt{2\pi}} \int_t^\infty e^{-x^2/2} dx} = \frac{\int_t^\infty x^4 e^{-x^2/2} dx}{\int_t^\infty e^{-x^2/2} dx} \leq 2t^4 \; . \tag{5}$$

We use Inequality (5) to upper bound (4):

$$\mathbb{E}\left[\sum_{i \in H} X_i^4\right] \leq O(m\alpha)\left(2 \cdot (4\ln^2(2/\alpha))^4\right) = O\big(m\alpha \log^8(2/\alpha)\big) \; .$$

By Markov's inequality, with probability at least $1 - 10^{-3}/3$, we have $\mathbb{E}\big[\sum_{i \in H} X_i^4\big] = O\big(m\alpha \log^8(2/\alpha)\big)$. Finally, since $X_1, \ldots, X_m$ are drawn i.i.d. from $\mathcal{N}(0,1)$, we have that $\max_{i \in [m]} |X_i| = O(\sqrt{\log m})$ with probability at least $1 - 10^{-3}/3$. $\qquad \square$

**Proposition A.5.** *Let $K_1 \geq 0$ and $K_2 \geq 0$. Let $a_1, \ldots, a_m \geq 0$ such that $\max_{i \in [m]} a_i^2 \leq K_1$ and $\sum_{i \in [m]} a_i^4 \leq K_2$. Let $X_1, \ldots, X_m$ be $m$ random variables drawn i.i.d. from $\mathcal{N}(0,1)$. Then, with probability at least $1 - 10^{-3} 12^{-d}$,*

$$\sum_{i \in [m]} a_i^2 X_i^2 = O\left(\sqrt{dK_2} + dK_1\right) \; .$$

*Proof.* Since $X_i \sim \mathcal{N}(0,1)$, the random variable $X_i^2$ is sub-exponential. Applying Bernstein's inequality for sub-exponential random variable [46, Theorem 2.8.2], for every $t \geq 0$,

$$\mathbf{Pr}\left[\sum_{i \in [m]} a_i X_i^2 \geq t\right] \leq \exp\left(-c\min\left(\frac{t^2}{\sum_{i \in [m]} a_i^4}, \frac{t}{\max_i a_i^2}\right)\right) \; , \tag{6}$$

where $c > 0$ is a universal constant.

We need to choose a value of $t$ such that the right-hand side of (6) is upper bounded by $10^{-3} 12^{-d}$. Given our assumptions on $a_1, \ldots, a_m$, it is sufficient to choose $t$ such that $t = \Omega\left(\sqrt{dK_2}\right)$ and $t = \Omega\left(dK_1\right)$. $\qquad \square$

**Proposition A.6.** *Let $\alpha \in (0,1)$. Let $X_1, \ldots, X_m$ be $m$ random variables drawn i.i.d. from $\mathcal{N}(0,1)$. With probability at least $1 - 10^{-3} 12^{-d}$, it holds that:*

$$\max_{L \subseteq [m]: |L| = \alpha m} \sum_{i \in L} X_i^2 = O(m\alpha \log(1/\alpha) + d) \; .$$

*Proof.* We define the threshold function

$$h_r(z) = \begin{cases} 0, & z \leq r \\ z, & z > r \end{cases}$$

with $r = 8\ln(1/\alpha)$. Since $z \leq r + h_r(z)$ for all $z > 0$,

$$\max_{L \subseteq [m], |L| = \alpha m} \sum_{i \in L} X_i^2 \leq \max_L \sum_{i \in L} r + \max_L \sum_{i \in L} h_r(X_i^2) \leq m\alpha \cdot r + \sum_{i \in [m]} h_r(X_i^2) \; .$$

We will use Chernoff-bound like arguments to obtain a high-probability upper bound on $\sum_{i \in [m]} h_r(X_i^2)$. For any $c > 0$ and $t > 0$, we have:

$$\mathbf{Pr}\left[\sum_{i \in [m]} h_r(X_i^2) \geq t\right] = \mathbf{Pr}\left[\exp\left(c \sum_{i \in [m]} h_r(X_i^2)\right) \geq \exp(c \cdot t)\right]$$

$$\leq e^{-ct} \mathbb{E}\left[\exp\left(c \sum_{i \in [m]} h_r(X_i^2)\right)\right] \tag{7}$$

$$= e^{-ct} \prod_{i \in [m]} \mathbb{E}\big[\exp\big(c \cdot h_r(X_i^2)\big)\big] \; ,$$

558 where the inequality follows from Markov's inequality.

559 Thus, it is sufficient to upper bound $\mathbb{E}\big[\exp(c \cdot h_r(X_i^2))\big]$. For any $c < 1/2$, we have

$$
\mathbb{E}\big[\exp\big(c \cdot h_r(X_i^2)\big)\big] = 1 \cdot \mathbf{Pr}\big[h_r(X_i^2) = 0\big] + \frac{1}{\sqrt{2\pi}} \int_{\sqrt{r}}^{\infty} e^{cx^2} e^{-x^2/2} dx
$$

$$
\leq 1 + \frac{1}{\sqrt{2\pi}\sqrt{1-2c}} \int_{\sqrt{r(1-2c)}}^{\infty} e^{-y^2/2} dy
$$

$$
= 1 + \frac{1}{\sqrt{1-2c}} \mathbf{Pr}\Big[X_i \geq \sqrt{r(1-2c)}\Big]
$$

$$
\leq 1 + \frac{1}{\sqrt{1-2c}} \exp\big(-r(\tfrac{1}{2} - c)\big) \ ,
$$

560 where the second inequality is obtained by substituting $x\sqrt{1-2c} = y$ in the integral, and the
561 last inequality uses the Gaussian tail bound that $\mathbf{Pr}[X_i \geq z] \leq e^{-z^2/2}$ for all $z > 0$. Recall that
562 $r = 8\ln(1/\alpha)$. We set $c = 1/4$, so that $\exp(-r/4) = \alpha^2$. Thus, we have that:

$$
\mathbb{E}\left[\exp\left(\frac{1}{4} \cdot h_r(X_i^2)\right)\right] \leq 1 + \sqrt{2}\alpha^2 \leq e^{\sqrt{2}\alpha^2} \tag{8}
$$

563 We substitute the upper bound (8) into (7) and obtain that

$$
\mathbf{Pr}\left[\sum_{i\in[m]} h_r(X_i^2) \geq t\right] \leq \exp\left(\frac{-t}{4} + \sqrt{2}m\alpha^2\right) \ .
$$

564 We can choose $t = 4\sqrt{2}m\alpha^2 + \Omega(d)$, and then conclude that with probability at least $1 - 10^{-3}12^{-d}$,

$$
\max_{L\subseteq[m]:|L|=\alpha m} \leq m\alpha r + \sum_{i\in[m]} h_r(X_i^2) = O(m\alpha r + m\alpha^2 + d) = O(m\alpha \log(1/\alpha) + d) \ .
$$

565 $\hfill\square$

566 **Lemma A.1.** *Let* $x \in \mathbb{R}^d$ *be the ground-truth vector. Fix* $\delta > 0$. *There exists constants* $\bar{\epsilon}(\delta)$ *and*
567 $c(\delta)$ *such that if we let* $0 < \epsilon \leq \bar{\epsilon}(\delta)$, *and we are given an* $\epsilon$-*corrupted set of* $n = \widetilde{\Omega}(c(\delta)d)$ *samples*
568 $(a_i, y_i)_{i\in[n]}$, *with probability at least* $0.99$, *for all* $w \in \Delta_{n,2\epsilon}$,

$$
\big\|Y_{G,w} - (I + 2xx^\top)\big\|_2 \leq \delta \ ,
$$

569 *where* $G$ *is the set of indices of the remaining good samples and* $Y_{G,w} = \sum_{i\in G} w_i y_i a_i a_i^\top$.

570 *Proof of Lemma A.1.* We recall the definition of $Y_{G,w} = \sum_{i\in G} w_i y_i a_i a_i^\top$. Let $\ell \leq \epsilon \cdot n$ and let
571 $\{(a_{n+i}, y_{n+i})\}_{i=1}^{\ell}$ be the set of samples that were removed by the $\epsilon$-corruption adversary. Let
572 $G' = G \cup \{n+1, \ldots, n+\ell\}$, $n' = n + \ell$, and $\epsilon' = \epsilon/(1+\epsilon)$. Note that without loss of generality,
573 we can assume that $|G| = (1 - \epsilon)n$ and $|G'| = (1 - \epsilon')n' = n$.

574 We define a mapping $\sigma : \Delta_{n,2\epsilon} \to \Delta_{n',3\epsilon'}$ such that

$$
\sigma(w)_i = \begin{cases} w_i & i \in [n] \\ 0 & \text{otherwise} \end{cases} . \tag{9}
$$

575 In other words, all the weights are the same for the samples with index in the set $[n]$, and are equal to
576 $0$ for the samples removed by the adversary. We can verify that $\sigma(w) \in \Delta_{n',3\epsilon'}$ for all $w \in \Delta_{n,2\epsilon}$
577 since $\sigma(w)_i \leq w_i \leq 1/(1-2\epsilon)n = 1/(1-3\epsilon')n'$ for all $i \in [n']$, and $\|\sigma(w)\|_1 = \|w\|_1 = 1$.
578 Furthermore, we have $Y_{G,w} = Y_{G',\sigma(w)}$ for all $w \in \Delta_{n,2\epsilon}$. We denote with $w^* \in \Delta_{n',3\epsilon'}$ the desired
579 uniform weighting of the samples with index in $G'$, i.e., $w_i^* = \frac{1}{(1-\epsilon')n'}\mathbb{1}_{i\in G'}$.

580 By triangle inequality, for any $w \in \Delta_{n,2\epsilon}$, it holds

$$
\big\|Y_{G',\sigma(w)} - (I + 2xx^\top)\big\|_2 \leq \big\|Y_{G',w^*} - (I + 2xx^\top)\big\|_2 + \big\|Y_{G',\sigma(w)-w^*}\big\|_2 , \tag{10}
$$

Thus, it suffices to show both $\left\|Y_{G',w^*} - (I + 2xx^\top)\right\|_2 \le \delta/2$ and $\left\|Y_{G',\sigma(w)-w^*}\right\|_2 \le \delta/2$. We upper bound the first term. By using the definition of $w^*$, note that

$$\left\|Y_{G',\sigma(w)} - (I + 2xx^\top)\right\|_2 = \left\|\sum_{i \in G'} w_i^* y_i a_i a_i^\top - (I + 2xx^\top)\right\|_2$$

$$= \left\|\sum_{i \in G'} \frac{1}{|G'|} y_i a_i a_i^\top - (I + 2xx^\top)\right\|_2 .$$

Since $\mathbb{E}\left[y_i a_i a_i^\top\right] = I + 2xx^\top$ for any $i \in G'$, we can use a concentration inequality to upper bound this term. By Lemma A.3, as long as $n \ge C(\delta/2) \cdot d \log d$, with probability at least $0.995$, we have

$$\left\|Y_{G',w^*} - (I + 2xx^\top)\right\|_2 \le \delta/2 . \tag{11}$$

It remains to show a high-probability upper bound to the second term $\left\|Y_{G',w^*-\sigma(w)}\right\|_2 \le \delta/2$ that holds for any $w \in \Delta_{w,2\epsilon}$. To achieve this goal, we will provide a high-probability upper bound to the following quantity:

$$J = \sup_{w \in \Delta_{n,2\epsilon}} \left\|Y_{G',w^*-\sigma(w)}\right\|_2 = \sup_{w \in \Delta_{n,2\epsilon}} \left\|\sum_{i \in G'} (w_i^* - \sigma(w)_i) y_i a_i a_i^\top\right\|_2 .$$

Note that for every $i$, the matrix $y_i a_i a_i^\top$ is positive semidefinite since $y_i \ge 0$. Thus, it holds that:

$$J \le \sup_{w \in \Delta_{n,2\epsilon}} \left\|\sum_{i \in G'} |w_i^* - \sigma(w)_i| y_i a_i a_i^\top\right\|_2 .$$

For any $w \in \Delta_{n,2\epsilon}$ and any $i \in [n']$, it is easy to see that $0 \le |w_i^* - \sigma(w)_i| \le \frac{1}{(1-2\epsilon)n}$. Additionally the weighting $w^*$ and $\sigma(w)$ cannot be too different. In particular, we can show the following upper bound:

$$\sum_{i \in G'} |w_i^* - \sigma(w)_i| \le \sum_{i=1}^{n'} |w_i^* - \sigma(w)_i| \le \sup_{w,w' \in \Delta_{n',3\epsilon'}} \sum_{i=1}^{n'} |w_i - w_i'| . \tag{12}$$

We observe that $\Delta_{n',3\epsilon'}$ can be seen as the convex combination of all possible uniform weighting over subsets of $n'(1 - 3\epsilon')$ samples. Thus, the maximum distance will be between two points of the convex hull, and we can upper bound (12) as:

$$\sum_{i \in G'} |w_i^* - \sigma(w)_i| \le \sup_{w,w' \in \Delta_{n',3\epsilon'}} \sum_{i=1}^{n'} |w_i - w_i'| \le \frac{6\epsilon' n}{n'(1 - 3\epsilon')} \le 6\epsilon . \tag{13}$$

Consider the family of weights defined as $\Gamma = \left\{\beta \in \mathbb{R}^{n'} : \sum_i \beta_i \le 6\epsilon \text{ and } 0 \le \beta_i \le \frac{1}{(1-2\epsilon)n}\right\}$. By the discussion above, we have that

$$J \le \sup_{w \in \Delta_{n,2\epsilon}} \left\|\sum_{i \in G'} |w_i^* - \sigma(w)_i| y_i a_i a_i^\top\right\|_2 \le \sup_{\beta \in \Gamma} \left\|\sum_{i \in G'} \beta_i y_i a_i a_i^\top\right\|_2 . \tag{14}$$

Since the map $\beta \mapsto \left\|\sum_{i \in G'} \beta_i y_i a_i a_i^\top\right\|_2$ is convex with respect to $\beta$, and $\Gamma$ is a convex set, the supremum over $\beta \in \Gamma$ of (14) is achieved at one of the extreme points of $\Gamma$. Thus, it holds:

$$J \le \sup_{\beta \in \Gamma} \left\|\sum_{i \in G'} \beta_i y_i a_i a_i^\top\right\|_2 \le \frac{1}{(1 - 2\epsilon)n} \max_{L \subseteq G', |L|=6\epsilon n} \left\|\sum_{i \in L} y_i a_i a_i^\top\right\|_2 .$$

For any vector $v \in \mathbb{S}^{d-1}$ in the unit sphere, let

$$J(v) = \max_{L \subseteq G', |L|=6\epsilon n} \sum_{i \in L} y_i (v^\top a_i)^2 ,$$

and note that $J \le \frac{1}{(1-2\epsilon)n} \sup_{v \in \mathbb{S}^{d-1}} J(v)$.

Without loss of generality, assume that $x = e_1$, where $e_1 \in \mathbb{R}^d$ is the first canonical vector. Given any vector $u \in \mathbb{R}^d$, we will denote with $u_1$ the first coordinate, and with $\widetilde{u} \in \mathbb{R}^{d-1}$ the remaining $d-1$ coordinates, i.e., $u = (u_1, \widetilde{u})$. Assuming $x = e_1$, we have that $y_i = (x^\top a_i)^2 = a_{i,1}^2$ for any $i \in G'$. Let $H = \{i \in G' : |a_{i,1}| \geq 4\ln^2(2/\epsilon)\}$. We consider a set $L \subseteq G'$ that always contains $H$ and then picks $6\epsilon n$ additional elements. In particular, it holds:

$$J(v) = \max_{L \subseteq G', |L| = 6\epsilon n} \sum_{i \in L} y_i(a_i^\top v_i)^2 \leq \left[ \sum_{i \in H} y_i(a_i^\top v_i)^2 + \max_{L \subseteq G' \setminus H, |L| = 6\epsilon n} \sum_{i \in L} y_i(a_i^\top v_i)^2 \right] . \quad (15)$$

The first term of the right-hand side of (15) can be rewritten as follows:

$$\sum_{i \in H} y_i(a_i^\top v)^2 = \sum_{i \in H} a_{i,1}^2 \left( \sum_{j=1}^d a_{i,j} v_j \right)^2 \leq 2 \sum_{i \in H} \left[ a_{i,1}^4 v_{i,1}^2 + a_{i,1}^2 (\widetilde{a}_i^\top \widetilde{v})^2 \right] . \quad (16)$$

For the second term of the right-hand side of (15), note that for any $i \in G' \setminus H$, it holds that $y_i = a_{i,1}^2 < 16\ln^4(2/\epsilon)$ due to the definition of $H$. Thus, we have that:

$$\max_{L \subseteq G' \setminus H, |L| = 6\epsilon n} \sum_{i \in L} y_i(a_i^\top v_i)^2 \leq 16\ln^4(2/\epsilon) \max_{L \subseteq G' \setminus H, |L| = 6\epsilon n} \sum_{i \in L} (a_i^\top v_i)^2 \quad (17)$$

$$\leq 32\ln^4(2/\epsilon) \max_{L \subseteq G' \setminus H, |L| = 6\epsilon n} \sum_{i \in L} \left[ a_i^2 v_1^2 + (\widetilde{v}^\top \widetilde{a}_i)^2 \right] . \quad (18)$$

Also, using the definition of $H$, note that for any $L \subseteq G' \setminus H$ with $|L| = 6\epsilon n$, we have that :

$$\sum_{i \in L} a_{i,1}^2 v_1^2 \leq \sum_{i \in L} a_{i,1}^2 = O\left(n\epsilon \ln^4(2/\epsilon)\right) . \quad (19)$$

By combining (16), (17), and (19) with (15), we obtain that:

$$J(v) = O\left( n\epsilon \log^8(1/\epsilon) + \sum_{i \in H} a_{i,1}^4 + \sum_{i \in H} a_{i,1}^2(\widetilde{a}_i^\top \widetilde{v})^2 + \max_{L \subseteq G' \setminus H, |L| = 6\epsilon n} \sum_{i \in L} (\widetilde{v}^\top \widetilde{a}_i)^2 \right) .$$

Let $E_1$ be the event of Proposition A.4 with $\alpha = \epsilon$ and $n = m$. That is, with probability at least $1 - 10^{-3}$, we have that $|H| = O(\epsilon n)$, $\sum_{i \in H} a_{i,1}^4 = O\left(n\epsilon \log^8(1/\epsilon)\right)$ and $\max_i a_{i,1}^2 = O(\log n)$. For the remaining of this proof, assume that $E_1$ is true, and thus:

$$J(v) = O\left( n\epsilon \log^8(1/\epsilon) + \sum_{i \in H} a_{i,1}^2(\widetilde{a}_i^\top \widetilde{v})^2 + \max_{L \subseteq G' \setminus H, |L| = 6\epsilon n} \sum_{i \in L} (\widetilde{v}^\top \widetilde{a}_i)^2 \right) . \quad (20)$$

Denote with $\overline{J}(v) = \sum_{i \in H} (\widetilde{a}_i^\top \widetilde{v})^2 + \max_{L \subseteq G' \setminus H, |L| = 6\epsilon n} \sum_{i \in L} (\widetilde{v}^\top \widetilde{a}_i)^2$ the last two terms of the right-hand side of (20). We will upper bound each term of $\overline{J}$ individually. First, observe that the random variables $Z_i = \widetilde{a}_i^\top \widetilde{v}/\|\widetilde{v}\|_2$ for $i \in G'$ are independent standard normal random variables. Let $E_2^v$ be the event of Proposition A.5 for the random variables $\{Z_i : i \in H\}$ and weights $\{a_{i,1}^2 : i \in H\}$ . That is, with probability at least $1 - 10^{-3}12^{-d}$, it holds that $\sum_{i \in H} a_{i,1}^2 Z_i^2 = O\left(\epsilon^{1/2}\sqrt{dn}\log^4(1/\epsilon) + d\log n\right)$. For the second term of $\overline{J}$, we can invoke Proposition A.6 over the random variables $\{Z_i : i \in G' \setminus H\}$ with $\alpha = 12\epsilon$ and $m = |G' \setminus H| \geq n/2$ (if $\epsilon < 1/2$). Let $E_3^v$ be the event of this proposition, that is, with probability at least $1 - 10^{-3}12^{-d}$, it holds that:

$$\max_{L \subseteq G' \setminus H, |L| = 6\epsilon n} \sum_{i \in L} (\widetilde{v}^\top \widetilde{a}_i)^2 = O(\epsilon n \log(1/\epsilon) + d) .$$

By taking a union bound of the events $E_2^v$ and $E_3^v$, we have that:

given $v$, with probability at least $1 - 2 \cdot 10^{-3}12^{-d}$,
$$\overline{J}(v) = O\left( \epsilon n \log(1/\epsilon) + \epsilon^{1/2}\sqrt{dn}\log^4(1/\epsilon) + d\log n \right). \quad (21)$$

623 Consider a $1/4$-net $\mathcal{N}$ of $\mathbb{S}^{d-1}$, where $|\mathcal{N}| = O(12^d)$. Note that for any $v \in \mathbb{S}^{d-1}$, it holds
624 $\sup_{v \in \mathbb{S}^{d-1}} \overline{J}(v) \leq 2 \sup_{v \in \mathcal{N}} \overline{J}(v)$. Thus, by taking a union bound over the event described in (21)
625 for all $v \in \mathcal{N}$, we obtain that with probability at least $1 - 2 \cdot 10^{-3}$, we have:

$$\sup_{v \in \mathbb{S}^{d-1}} \overline{J}(v) = O\Big(\epsilon n \log(1/\epsilon) + d \log n + \sqrt{dn\epsilon} \log^2(1/\epsilon)\Big) \tag{22}$$

626 We finally combine (22) and (20) to conclude that with probability at least $1 - 0.995$, it holds that

$$J \leq O\Big(\frac{1}{n}\Big[d \log n + \epsilon^{1/2}\sqrt{dn} \log^4(1/\epsilon) + n\sqrt{\epsilon} \log^8(1/\epsilon)\Big]\Big) \ .$$

627 We can pick a sufficiently small $\epsilon$ (depending only on $\delta$) and $n \geq c(\delta)d \log d$ for a sufficiently large
628 constant $c(\delta)$ so that with probability at least $1 - 0.995$, it holds that $J \leq \delta/2$. Utilizing this result
629 along with (11) to upper bound (10) yields the desired statement. $\qquad\square$

## B  Omitted Proofs in Section 5

631 **Lemma 5.1.** *Let $x \in \mathbb{R}^d$ be the ground-truth vector. Let $\mathcal{D}_z$ be the distribution of gradients at $z$ as*
632 *defined in Equation (2). For any $z$ with $\|z - x\|_2 \leq 1$, the covariance matrix $\Sigma_z$ of $\mathcal{D}_z$ satisfies*

$$\Sigma_z \preceq O\Big(\|z - x\|_2^2\Big) I \ .$$

633 *Proof of Lemma 5.1.* Recall that $g \sim \mathcal{D}_z$ is distributed as

$$g = \frac{\partial}{\partial z}\left[\Big(\langle a, z \rangle^2 - \langle a, x \rangle^2\Big)^2\right] = 4\Big(\langle a, z \rangle^2 - \langle a, x \rangle^2\Big)\langle a, z \rangle a \quad \text{where} \quad a \sim \mathcal{N}(0, I) \ .$$

634 Let $\mu_z = \mathbb{E}_{g \sim \mathcal{D}_z}[g]$. We have

$$0 \preceq \Sigma_z = \mathop{\mathbb{E}}_{g \sim \mathcal{D}_z}\big[gg^\top\big] - \mu_z \mu_z^\top \preceq \mathop{\mathbb{E}}_{g \sim \mathcal{D}_z}\big[gg^\top\big] \ .$$

635 Consequently, it suffices to upper bound the spectral norm of $\mathbb{E}_{g \sim \mathcal{D}_z}\big[gg^\top\big]$. Let $h = z - x$.

$$\begin{aligned}
\|\Sigma_z\|_2 &\leq \left\|\mathop{\mathbb{E}}_{g \sim \mathcal{D}_z}\big[gg^\top\big]\right\|_2 \\
&= \max_{\|v\|_2=1} v^\top \mathop{\mathbb{E}}_{g \sim \mathcal{D}_z}\big[gg^\top\big] v \\
&= \max_{\|v\|_2=1} \mathop{\mathbb{E}}_{g \sim \mathcal{D}_z}\big[\langle g, v \rangle^2\big] \\
&= 16 \max_{\|v\|_2=1} \mathop{\mathbb{E}}_{a \sim \mathcal{N}(0,I)}\left[\Big(\big(\langle a, z \rangle^2 - \langle a, x \rangle^2\big)\langle a, z \rangle \langle a, v \rangle\Big)^2\right] \\
&= 16 \max_{\|v\|_2=1} \mathop{\mathbb{E}}_{a \sim \mathcal{N}(0,I)}\left[\Big(\big(\langle a, h \rangle^2 + 2\langle a, x \rangle \langle a, h \rangle\big)\langle a, x + h \rangle \langle a, v \rangle\Big)^2\right] \\
&= 16 \max_{\|v\|_2=1} \mathop{\mathbb{E}}_{a \sim \mathcal{N}(0,I)}\left[\langle a, h \rangle^2 \langle a, 2x + h \rangle^2 \langle a, x + h \rangle^2 \langle a, v \rangle^2\right] \\
&\leq 16 \max_{\|v\|_2=1} \left(\mathop{\mathbb{E}}_a\big[\langle a, h \rangle^8\big] \mathop{\mathbb{E}}_a\big[\langle a, 2x + h \rangle^8\big] \mathop{\mathbb{E}}_a\big[\langle a, x + h \rangle^8\big] \mathop{\mathbb{E}}_a\big[\langle a, v \rangle^8\big]\right)^{1/4} \\
&= 16 \max_{\|v\|_2=1} \left(105^4 \|h\|_2^8 \|2x + h\|_2^8 \|x + h\|_2^8 \|v\|_2^8\right)^{1/4} \\
&= (16 \cdot 105) \|h\|_2^2 \|2x + h\|_2^2 \|x + h\|_2^2 \\
&= O(\|h\|_2^2) \ .
\end{aligned}$$

636 The last inequality follows from the Cauchy-Schwarz inequality. The last step uses the fact that
637 $\|2x + h\|_2 = O(1)$ and $\|x + h\|_2 = O(1)$, which follows from $\|x\|_2 = 1$ and $\|h\|_2 \leq 1$. $\qquad\square$

**Lemma 5.3.** *Let $x \in \mathbb{R}^d$ be the ground-truth vector. Suppose at iteration $t$ of Algorithm 2, the current solution $z_t$ satisfies $\|z_t - x\|_2 \leq \frac{1}{8}$. Let $\mu_{z_t}$ denote the expected true gradient at $z_t$ defined in Equation* (3). *Suppose the estimated gradient $\widehat{\mu}_{z_t} \in \mathbb{R}^d$ satisfies $\|\widehat{\mu}_{z_t} - \mu_{z_t}\|_2 \leq c\,\|z_t - x\|_2$ for $c = 4$. Then, we have*

$$\|z_{t+1} - x\|_2^2 \leq 0.99\,\|z_t - x\|_2^2 \ .$$

*Proof of Lemma 5.3.* Recall that $g \sim \mathcal{D}_z$ is distributed as

$$g = \frac{\partial}{\partial z}\left[\left(\langle a, z\rangle^2 - \langle a, x\rangle^2\right)^2\right] = 4\left(\langle a, z\rangle^2 - \langle a, x\rangle^2\right)\langle a, z\rangle a \quad \text{where} \quad a \sim \mathcal{N}(0, I) \ .$$

We can compute the mean $\mu_z$ of $\mathcal{D}_z$ using moments of Gaussian:

$$\mu_z = \mathop{\mathbb{E}}_{g \sim \mathcal{D}_z}[g] = \left(12\,\|z\|_2^2 - 4\,\|x\|_2^2\right) z - 8\,\langle x, z\rangle\, x \ .$$

Consider one step of gradient descent in Algorithm 2: $z_{t+1} = z_t - \eta\widehat{\mu}_{z_t}$, where $\widehat{\mu}_{z_t}$ is close to $\mu_z$. We have

$$\|z_{t+1} - x\|_2^2 = \|z_t - \eta\widehat{\mu}_{z_t} - x\|_2^2 = \|z_t - x\|_2^2 - 2\eta\,\langle\widehat{\mu}_{z_t}, z_t - x\rangle + \eta^2\,\langle\widehat{\mu}_{z_t}, \widehat{\mu}_{z_t}\rangle$$

To prove convergence, we need to lower bound $\langle\widehat{\mu}_{z_t}, z_t - x\rangle$ and upper bound $\langle\widehat{\mu}_{z_t}, \widehat{\mu}_{z_t}\rangle$.

Let $z = z_t$ and $h = z - x$. Substituting $z = x + h$ in the expression for $\mu_z$, we get:

$$\mu_z = \left(12\,\|x + h\|_2^2 - 4\,\|x\|_2^2\right)(x + h) - 8\,\langle x, x + h\rangle\, x$$
$$= \left(16\,\langle x, h\rangle + 12\,\|h\|_2^2\right) x + \left(8\,\|x\|_2^2 + 24\,\langle x, h\rangle + 12\,\|h\|_2^2\right) h \ .$$

We will be using the assumptions of this lemma: $\|x\|_2 = 1$, $\|h\|_2 \leq \frac{1}{8}$, and $\|\widehat{\mu}_z - \mu_z\|_2 \leq c\,\|h\|_2$.

First we lower bound $\langle\widehat{\mu}_z, h\rangle$.

$$\langle\widehat{\mu}_z, h\rangle = \langle\mu_z, h\rangle + \langle\widehat{\mu}_z - \mu_z, h\rangle$$
$$= 16\,\langle x, h\rangle^2 + 36\,\langle x, h\rangle\,\|h\|_2^2 + 8\,\|x\|_2^2\,\|h\|_2^2 + 12\,\|h\|_2^4 + \langle\widehat{\mu}_z - \mu_z, h\rangle$$
$$\geq -\tfrac{81}{4}\,\|h\|_2^4 + 8\,\|x\|_2^2\,\|h\|_2^2 + 12\,\|h\|_2^4 - c\,\|h\|_2^2$$
$$\geq \left(-\tfrac{81}{256} + 8 + \tfrac{12}{64} - c\right)\|h\|_2^2$$
$$\geq (7.5 - c)\,\|h\|_2^2 \ .$$

The first inequality uses the fact that $16\,\langle x, h\rangle^2 + 36\,\langle x, h\rangle\,\|h\|_2^2$ is a quadratic function of $\langle x, h\rangle$, which has minimum value $-\tfrac{81}{4}\,\|h\|_2^4$ for all $\langle x, h\rangle \in \mathbb{R}$.

Next we upper bound $\|\widehat{\mu}_z\|_2$ using the triangle inequality.

$$\|\widehat{\mu}_z\|_2 \leq \|\mu_z\|_2 + \|\widehat{\mu}_z - \mu_z\|_2$$
$$\leq \left(16\,\langle x, h\rangle + 12\,\|h\|_2^2\right)\|x\|_2 + \left(8\,\|x\|_2^2 + 24\,\langle x, h\rangle + 12\,\|h\|_2^2\right)\|h\|_2 + c\,\|h\|_2$$
$$\leq \left(16 + \tfrac{12}{8} + 8 + \tfrac{24}{8} + \tfrac{12}{64} + c\right)\|h\|_2$$
$$\leq (29 + c)\,\|h\|_2 \ .$$

Putting everything together, we have

$$\|z_{t+1} - x\|_2^2 = \|z_t - x\|_2^2 - 2\eta\,\langle\widehat{\mu}_{z_t}, z_t - x\rangle + \eta^2\,\langle\widehat{\mu}_{z_t}, \widehat{\mu}_{z_t}\rangle$$
$$\leq \left[1 - 2(7.5 - c)\eta + (29 + c)^2\eta^2\right]\|z_t - x\|_2^2 \ .$$

Choosing $c = 4$ and $\eta = 1/300$ gives that $\|z_{t+1} - x\|_2^2 \leq 0.99\,\|z_t - x\|_2^2$. $\qquad\square$

## C   A Counter-Example for Previous Algorithms

In this work, we consider a general setting that allows adversarial corruption in both the measurement vectors $a_i$'s and the intensity measurements $y_i$'s. Prior work in robust phase retrieval has addressed a special case where the adversarial corruption is restricted to the $y_i$'s, still assuming that the measuring vectors $a_i$'s are independently sampled from the Gaussian distribution [27, 51]. In this section, we construct a counter-example demonstrating the failure of algorithms developed for the restricted corruption setting when applied to the more general setting considered in our paper.

The Median Truncated Wirtinger Flow Algorithm [51] is an algorithm to address the robust phase retrieval problem with adversarial corruption limited to the $y_i$'s. The algorithm first initializes $z^{(0)}$ using the spectral method. Let $\alpha \geq 3$. In particular, $z^{(0)}$ is computed as the top eigenvector of the empirical matrix $Y := \frac{1}{m} \sum_{i=1}^{m} y_i a_i a_i^\top \mathbb{1}_{|y_i| \leq \alpha^2 \operatorname{med}(\{y_i\}_{i=1}^m)}$ that only uses a truncated set of samples, where the threshold is determined by $\operatorname{med}(\{y_i\}_{i=1}^m)$, the median over all $y_i$'s. The analysis of the algorithm relies on the fact that as long as the fraction of outliers is not too large and the sample complexity is large enough, the initialization is guaranteed to be within a small neighborhood of the ground truth.

We show that this initialization can fail to remove the distortion introduced by the adversarial if corruption is allowed for both $a_i$'s and $y_i$'s. Let $x \in \mathbb{S}^{d-1}$ be the ground truth unit vector. We construct an $\epsilon$-corruption adversary that can manipulate the top eigenvector of the empirical covariance matrix $Y = \sum_{i=1}^{n} y_i a_i a_i^\top$, even when all $y_i$'s are accurately calculated as $y_i = (a_i^\top x)^2$.

Let $u \in \mathbb{S}^{d-1}$ be a unit vector such that $x^\top u = 0$. Suppose the adversary changes $1\%$ of the $a_i$'s to $a_i = \sqrt{d - 1/25} \cdot u + (1/5) \cdot x$, and suppose that all the $y_i$'s are accurate. In particular, the length of the corrupted $a_i$'s is comparable to the length of a random Gaussian vector, and the corresponding intensity measurements satisfy $y_i = (a_i^\top x)^2 = 1/25$. Let $z = (a^\top x)^2$ for a random vector $a \sim \mathcal{N}(0, I)$. By direct computation, note that $\mathbf{Pr}[z \geq 0.2] \geq 0.6$. Thus, with high-constant probability, the median-truncated initialization in [51] is not able to filter out any of those samples. However, after the adversarial corruption, the top eigenvector of $\mathbb{E}\left[\sum_{i=1}^{n} y_i a_i a_i^\top\right] \approx O(d)uu^\top + O(\sqrt{d})(ux^\top + xu^\top) + O(1)(I + 2xx^\top)$ will be manipulated to $u$, which is far from the ground truth $x$.

