# OpenReview forum: "Outlier-Robust Phase Retrieval in Nearly-Linear Time"
_NeurIPS.cc/2025/Conference — Submitted to NeurIPS 2025_

### Official Review · Reviewer_cUwv · 2025-06-26

**Clarity:** 2
**Significance:** 3
**Originality:** 2
**Rating:** 3
**Confidence:** 4

**Summary:**

This paper presents a computationally efficient, nearly linear-time algorithm for solving the real-valued robust phase recovery problem, which aims to reconstruct a signal from phaseless measurements corrupted by adversarial outliers. The proposed method consists of two key stages: (1) a robust spectral initialization via a lightweight convex program to obtain a rough estimate near the ground truth, followed by (2) a refinement step using a robust gradient descent variant to achieve exact recovery within specified accuracy. The authors demonstrate that their approach is both near-sample-optimal and computationally efficient, with potential applicability to other non-convex optimization problems. While the paper provides theoretical analysis of the algorithm's superiority, some concerns were raised about insufficient mathematical rigor (e.g., unproven lemmas) and lack of numerical simulations, casting doubt on whether it currently meets NeurIPS publication standards. The framework is noted for its conceptual simplicity and generalizability, but may require additional theoretical and empirical validation.

**Questions:**

1. Why not conduct numerical simulation to compare with the existing algorithms and verify the impact of different attack strategies on the proposed algorithm?
2. Why does the article extensively use Big O notation instead of accurately writing out the specific constants?
3. Why is the step size of gradient descent in Algorithm 2 set to $1/300$?
4. In line 127, "the corrupted samples can only add directions to Yw but cannot remove directions." What is the meaning of "direction"? Why do corrupted samples add directions to Y_{w}?
5. Why do you compute w that minimizes the sum of the largest "two" eigenvalues of Y_{w}?

**Ethical Concerns:**

["NO or VERY MINOR ethics concerns only"]

**Final Justification:**

I have carefully reviewed the authors’ replies, and I would like to retain my original score.

**Limitations:**

This article is not only not clearly expressed in theory, but also lacks numerical simulation experiments.

**Quality:**

2

**Strengths And Weaknesses:**

Strengths: Compared with existing works, the main innovations of this article are as follows:
1. Phase retrieval is a well-studied non-convex problem, and analyzing the performance of robust algorithms on it remains crucial.
2. This article proposes a new low-complexity algorithm for robust phase recovery through robust spectral initialization and robust gradient descent.
3. This article demonstrates that the proposed algorithm can approximately solve the robust phase recovery problem within a linear time of the input size, under any given accuracy condition.

Weaknesses:
1. The paper does not explicitly point out any difficulties in terms of theory or technology that the theoretical analysis and algorithm derivation of the proposed algorithm compared with the existing works.
2. This paper lacks numerical simulation experiments, which prevents it from demonstrating the superiority of the proposed algorithm compared to other algorithms.
3. The descriptions of some lemmas in this paper are rather vague, for example:
3.1. The description of the parameter $\epsilon$ in Lemma 3.1 as "sufficiently small"
3.2. The description of the parameter $\delta$ in Lemma 4.1 is insufficient
3.3. The constant corresponding to $O\left(\left\|z-x\right\|_{2}^{2}\right)$ in Lemma 5.1 is not given
4. The proofs of some theorems in this article are not rigorous, for example:
4.1. In the proof of Lemma 3.1, on line 254 of the formula, the subtraction of two $O(\delta)$ terms results in $0$, which is incorrect
4.2. In the proof of Lemma 5.2, on line 300, the second inequality selects $\epsilon(c)=O(c^{2})$, which may not satisfy the condition $\epsilon(c)=\epsilon_{3}/2$

---

> ### Author Rebuttal · Authors · 2025-07-31
>
> We thank the reviewer for their time spent reading our work.
>
>
> ---
>
> **Reviewer**: The paper does not explicitly point out any difficulties in terms of theory or technology that the theoretical analysis and algorithm derivation of the proposed algorithm compared with the existing works.
>
> **Authors**:  A key technical contribution of our paper is the robust spectral initialization step, which differs significantly from prior work. Specifically, we design a nonnegative reweighting scheme that ensures the empirical covariance matrix is robust to outliers without explicitly computing the covariance matrix, which would require matrix multiplication time $O(d^{2.37})$. Instead, we introduce a new Ky Fan norm formulation and show that good weights can be computed in nearly-linear time $\tilde O(nd) \approx \tilde O(d^2)$. Note that concurrent work [BR25] (as discussed in Lines 69–71) performs robust spectral initialization in a different way but is much slower than linear time.
>
> ---
>
> **Reviewer**: This paper lacks numerical simulation experiments, which prevents it from demonstrating the superiority of the proposed algorithm compared to other algorithms.
>
> **Authors**: We believe our results are substantial even without experiments, and we hope that our theoretical results are judged based on their merits.
>
> ---
>
> **Reviewer**: The descriptions of some lemmas in this paper are rather vague, for example: 3.1. The description of the parameter  in Lemma 3.1 as "sufficiently small" 3.2. The description of the parameter  in Lemma 4.1 is insufficient 3.3. The constant corresponding to $O(|z - x|^2_2)$ in Lemma 5.1 is not given.
>
> **Authors**: We respectfully disagree. Our theorems, lemmas, and proofs are rigorous. A key point is that many constants appear throughout our analysis, and explicitly tracking each of them would introduce notational overhead without providing additional insight. Omitting explicit constants is standard practice in the theory community. As a concrete example, in Appendix B where we prove Lemma 5.1, the constant can be written explicitly as $151 > 16 \cdot 105 \cdot (1/8)^2 \cdot (2+1/8)^2 \cdot (1+1/8)^2$. While it is possible to include this constant in Lemma 5.1 and substitute it into other parts of our proofs, doing so would make the presentation unnecessarily cumbersome without affecting the correctness of our results.
>
>
> ---
>
> **Reviewer**: The proofs of some theorems in this article are not rigorous (4.1 - Example in line 254; 4-2,  Proof of Lemma 5.2)
>
> **Authors**: Thank you for pointing this out. The statement can indeed be made more precise here. In Lemma 4.1, we should have stated the inequalities of Line 248 using a value u: e.g., the top eigenvalue is between $3-u$ and $3+u$ for some $u = O(\delta)$, and complete the rest of the proof using $u$.
>
> We should have stated in Line 254 that $(3+u)\alpha_1^2 + (1+u)(1-\alpha_1^2) \le 1 + O(\delta) + 2\alpha_1^2$. We will revise the paper accordingly.
>
> ---
>
> **Reviewer**: Why is the step size of GD in algorithm 2 set to 1/300?
>
> **Authors**: Our choice of step size $\eta = 1/300$ is close to the best-possible constant to minimize the expression in Line 654 when $c = 4$.
>
> ---
>
> **Reviewer**: In line 127, "the corrupted samples can only add directions to Yw but cannot remove directions." What is the meaning of "direction"? Why do corrupted samples add directions to Y_{w}?
>
> **Authors**: This was explained in Lines 122-126. When there is no corruption, we have $\mathbb{E}[Y_w] = I + 2 x x^\top$, which is a matrix with one eigenvalue $3$ (in the direction of $\textbf{x}$) and all other eigenvalues $1$. Corrupted samples may perturb this spectrum, but since each term $y_i \textbf{a}_i \textbf{a}_i^\top$ is a positive semidefinite matrix, they cannot reduce variance in any direction (that is, the corrupted samples cannot decrease $v^\top Y_w v$ for any direction $v$). We will clarify this.
>
> ---
>
> **Reviewer**: Why do you compute w that minimizes the sum of the largest "two" eigenvalues of Y_{w}?
>
> **Authors**: Suppose the adversarial corrupts $Y_w$ to become $M = I + 2 x x^\top + 2.01 y y^\top$ for some arbitrary irrelevant direction $y$. The largest eigenvalue of $M$ is close to $3$, which is the same as what we expect when there is no corruption, but the top eigenvector of $M$ is not a good initialization for converging to the ground truth $x$.

---

> > ### Comment · Reviewer_cUwv · 2025-08-08
> >
> > Thank you for the response. I would like to retain my original score.

---

### Official Review · Reviewer_6VUx · 2025-07-03

**Clarity:** 3
**Significance:** 2
**Originality:** 2
**Rating:** 2
**Confidence:** 5

**Summary:**

This article proposes a two-stage algorithm for the real-valued phase retrieval problem, with a small fraction of measurement pair corrupted. The algorithm first estimates a high-quality initial point through a spectral-based convex optimization that adaptively weights corrupted sample matrices to approximate the ideal matrix. This initialization is then refined through corruption-resistant gradient descent using robust mean estimation. The paper claims its near-optimal sample complexity and near-linear runtime with theoretical guarantees.

**Questions:**

1.	The main contribution claimed is the near-linear runtime. However, the analysis about lemma 4.1 and 5.2 shows that the running time depends on the reciprocal of epsilon and desired precision. Since they are rather small, it's unclear whether the method is more efficient over other alternatives. How its practical runtime compared with similar approaches (e.g., [3])?

**Ethical Concerns:**

["NO or VERY MINOR ethics concerns only"]

**Final Justification:**

I remain unconvinced by the submission. The rebuttal was not informative and failed to meaningfully address the core concerns raised in the initial review. In particular, the key issues regarding the practical relevance of the proposed method and the lack of necessary experimental evidence remain unresolved. Moreover, they appear reluctant to acknowledge these limitations or outline concrete plans to address them in future work. This lack of engagement further undermines confidence in the contribution.

As such, the work falls short in both depth and rigor, and the central claims remain insufficiently supported. I therefore downgrade my rating to a rejection.

**Limitations:**

No limitation is discussed.

**Paper Formatting Concerns:**

NIL

**Quality:**

2

**Strengths And Weaknesses:**

**Strength**:

1.	Paper is well-written and provides a clear picture of the setting of the problem

2.	Phase retrieval is a non-convex problem with important applications, studying its robustness certainly is an important

3.	the algorithm is weill supported by the theoretical analysis.

**Weakness**

1.  I do not believe the problem setting studied in this paper has genuine practical value. In phase retrieval, if noise is considered, then all pairs will be corrupted. If outliers are considered, they would correspond to measurements with very large noise levels. The setting proposed in the paper does not make sense in real-world scenarios.

2. The analysis appears to be more incremental in nature compared to prior theoretical works on phase retrieval. Specifically, the main novelty of the spectral initialization step lies in assigning a nonnegative weight to each sample. For the gradient descent step, the problem is essentially reduced to the analysis of robust mean estimation algorithms.

3. Although the authors claim it is a theoretical paper, the work essentially proposes an algorithm with theoretical justification. The technique in the theoretical derivations themselves have no novelty. Since the primary objective of the paper is to develop a robust algorithm, it is quite surprising that no experimental evaluation is provided to assess its performance or efficiency. For an algorithm study paper, the absence of empirical validation is unacceptable.

This paper is a resubmission of the work originally submitted to NeurIPS 2024. One of the main concerns previously raised was the lack of numerical experiments; however, this revision fails to address that issue entirely.

---

> ### Author Rebuttal · Authors · 2025-07-31
>
> We thank the reviewer for their time and thoughtful comments.
>
> ---
>
> **Reviewer**:  Although the authors claim it is a theoretical paper, the work essentially proposes an algorithm with theoretical justification. The techniques in the theoretical derivations themselves have no novelty.
>
> **Authors**: We respectfully disagree. Our contribution is both algorithmic and theoretical, and the techniques we introduce are novel in the context of robust phase retrieval. In particular, our main result establishes that one can solve phase retrieval with arbitrary corruption on the pairs $(\textbf{a}_i, y_i)$ in nearly-linear time and with near-optimal sample complexity. Prior work typically considers simpler corruption models, such as adversarial perturbations limited to the intensity measurements $y_i$. Handling corruptions in both $(\textbf{a}_i, y_i)$ presents new challenges, which we address through novel techniques.
> A key technical contribution of our paper is the robust spectral initialization step, which differs significantly from prior work. Specifically, we design a nonnegative reweighting scheme that ensures the empirical covariance matrix is robust to outliers without explicitly computing the covariance matrix, which would require matrix multiplication time $O(d^{2.37})$. Instead, we introduce a new Ky Fan norm formulation and show that good weights can be computed in nearly-linear time $\tilde O(nd) \approx \tilde O(d^2)$. Note that concurrent work [BR25] (as discussed in Lines 69–71) performs robust spectral initialization in a different way but is much slower than linear time. To our knowledge, no prior work achieves this level of robustness and efficiency; we would welcome the reviewer pointing us to any such existing result.
>
> ---
>
> **Reviewer**: The analysis appears to be more incremental in nature compared to prior theoretical works on phase retrieval. Specifically, the main novelty of the spectral initialization step lies in assigning a nonnegative weight to each sample.
>
> **Authors**: The main novelty is that we can compute good weights in nearly-linear time. We invite the reviewer to point to prior work that achieves this when there is corruption in both $a_i$ and $y_i$.
> As noted above, our spectral initialization is not merely a reweighting heuristic. It is backed by a rigorous analysis showing how to select these weights efficiently via Ky Fan norm minimization, and how this leads to accurate recovery despite arbitrary corruptions. This represents a departure from existing techniques and is not a straightforward extension of prior methods.
>
> ---
>
> **Reviewer**: This paper is a resubmission of the work originally submitted to NeurIPS 2024. One of the main concerns previously raised was the lack of numerical experiments; however, this revision fails to address that issue entirely.
>
> **Authors**: We believe our results are substantial even without experiments, and we hope that our theoretical results are judged based on their merits.
>
> That said, the current version has significantly improved over the NeurIPS24 submission: we have clarified the setting, strengthened the presentation of the results, and expanded the discussion of the techniques.
>
> ---
>
> **Reviewer** : I do not believe the problem setting studied in this paper has genuine practical value. In phase retrieval, if noise is considered, then all pairs will be corrupted. If outliers are considered, they would correspond to measurements with very large noise levels. The setting proposed in the paper does not make sense in real-world scenarios.
>
> **Authors** : We respectfully disagree. We welcome the study of other adversarial models. We want to emphasize that there is value in studying phase retrieval in the strong contamination model we consider in this paper. Our results show that even against such a strong adversary, phase retrieval can be solved with only logarithmic overhead in both sample and computational complexity. Extending our work to the setting with noisy observations is an exciting avenue of future work, where our nearly linear-time robust initialization step will remain useful.

---

> > ### Comment · Reviewer_6VUx · 2025-08-05
> >
> > The rebuttal is not informative. The main concerns about practical relevance and the absence of experiments remain unaddressed. The authors offer assertions without providing substantial supporting evidence.

---

### Official Review · Reviewer_WB9r · 2025-07-03

**Clarity:** 3
**Significance:** 3
**Originality:** 3
**Rating:** 5
**Confidence:** 4

**Summary:**

This paper studies phase retrieval under the strong contamination model, where an adversary can arbitrarily replace an $\epsilon$-fraction of measurements $y_i = \langle a_i, x\rangle^2$ and the corresponding measurement vectors $a_i$. It proposes suitable modifications to the standard 2-step approach (spectral initialization followed by gradient descent) that lead to an outlier-robust algorithm with near-optimal runtime $\tilde{O}(nd)$ and sample complexity $O(d \cdot \mathrm{polylog}(1 / \Delta))$, where $\Delta$ is the target estimation accuracy. The first stage -- robust spectral initilization -- reduces to finding per-sample weights $w_i$ such that

$$
\widehat{Y}_{w} := \sum_{i} w_i y_i a_i a_i^{\top} \approx I + 2 xx^{\top}
$$

by means of solving a generalized packing SDP proposed in prior work. The latter stage -- robust gradient descent -- leverages the fact that it is sufficient to obtain an estimate of the gradient, $\widehat{g}(x_k)$, satisfying the aiming inequality

$$
\langle \widehat{g}(x_k), x_k - x\rangle \gtrsim \|x_k - x\|^2
$$

in a neighborhood of the solution set.

**Questions:**

My main questions are the following:

1. Appendix C shows that prior "robust" approaches -- focusing on corruption solely in the measurement space -- fail when measurement vectors can be corrupted. However, my understanding is that the argument focuses on the failure of spectral initialization. Is it possible that, given a sufficiently good initial estimate, prior methods (such as optimizing the robust $\ell_1$ loss a-la Duchi-Ruan) succeed in recovering $x$? In my opinion, a formal argument ruling this out would considerably strengthen this paper.
2. Is it possible to remove the sample-splitting requirement by following an approach similar to the SEVER algorithm (https://proceedings.mlr.press/v97/diakonikolas19a.html), wherein samples are used to certify first-order stationarity and discarded iteratively based on outlier scores? See, also, the recent work by Gao et al. (https://arxiv.org/abs/2412.11003).

Other minor questions:
- What is the relationship between $\epsilon(\delta)$ in Lemma 4.1 and the upper bound $\epsilon_0$ on the contamination level?
- How much of the overall analysis depends on Gaussianity? I suspect the robust gradient descent analysis can be easily extended.
- Can you expand on how you anticipate attacking the sample-splitting requirement? I suspect that verifying stability conditions without fresh samples is very difficult in general settings.
- Is there a weaker notion of adversary suitable for some of the corruptions you describe in the introduction (hardware noise + miscalibration)?

**Ethical Concerns:**

["NO or VERY MINOR ethics concerns only"]

**Final Justification:**

I did not have any major reservations about this paper, so I am maintaining my original score.

**Limitations:**

Yes

**Quality:**

3

**Strengths And Weaknesses:**

Strengths:
- The overall algorithm follows the standard 2-step scheme and is simple to state (although the solver for the generalized packing SDP might be complicated).
- The runtime is near-optimal and the sample complexity is linear in $d$ for moderate estimation accuracies $\Delta$.

Weaknesses:
- Real gaussian measurement vectors are somewhat unrealistic in phase retrieval.
- The algorithm requires a fresh batch of $d \log(1 / \Delta)$ samples in each iteration.
- I am not sure if the strong contamination model is the right way to study corruptions to the measurement vectors (other than, of course, adversarial tampering). For example, neither hardware noise nor miscalibration seem to require an adversary that is allowed to inspect all inputs before deciding which ones to corrupt (and how to corrupt them).

---

> ### Author Rebuttal · Authors · 2025-07-31
>
> We thank the reviewer for taking the time to read our work and provide their feedback.
>
> ---
>
> **Reviewer**: Is it possible to remove the sample-splitting requirement by following an approach similar to the SEVER algorithm (https://proceedings.mlr.press/v97/diakonikolas19a.html), Can you expand on how you anticipate attacking the sample-splitting requirement? I suspect that verifying stability conditions without fresh samples is very difficult in general settings.
>
> **Authors**: This is a good observation. We believe that a covering argument on the stability condition would allow us to remove the sample-splitting (and thus the dependency on $\Delta$). At a high level, proving an $\epsilon$-net plus union bound argument for the stability conditions (i.e., stability of the mean and covariance of individual gradients) to hold uniformly for all $z$ is not straightforward. These conditions only hold when none of the $a_i$'s are close to the current solution $\pm z$. Of course, if the $a_i$'s are chosen **after** $z$, this holds with high-probability, and this is why we use a fresh set of samples in each iteration. One possible idea is to consider an $\epsilon$-net argument in regions far from all $a_i$'s, and argue that gradient descent never gets close to any of the $a_i$'s. We defer this to future work, as our main technical contribution is in the robust initialization step (via the Ky Fan SDP solver).
>
> ---
>
> **Reviewer**: What is the relationship between $\epsilon(\delta)$  in Lemma 4.1 and the upper bound $\epsilon_0$ on the contamination level?
>
> **Authors**: Thank you for asking this. We need Lemma 4.1 to hold for our choice of $\delta$ (from Line 256), and the simplest way to do this is to require $\epsilon_0 \le \epsilon(\delta)$. We will clarify this.
>
> ---
>
> **Reviewer**: Is there a weaker notion of adversary suitable for some of the corruptions you describe in the introduction (hardware noise + miscalibration)?
>
> **Authors**: We welcome the study of other adversarial models. We want to emphasize that there is value in studying phase retrieval in the strong contamination model we consider in this paper. Our results show that even against such a strong adversary, phase retrieval can be solved with only logarithmic overhead in both sample and computational complexity.
>
> ---
>
> **Reviewer**: Appendix C shows that prior "robust" approaches -- focusing on corruption solely in the measurement space -- fail when measurement vectors can be corrupted. However, my understanding is that the argument focuses on the failure of spectral initialization. Is it possible that, given a sufficiently good initial estimate, prior methods (such as optimizing the robust  loss a-la Duchi-Ruan) succeed in recovering ? In my opinion, a formal argument ruling this out would considerably strengthen this paper.
>
> **Authors**: We currently do not have such a formal impossibility result. We agree that proving this formally would strengthen the paper and view it as a promising direction for future work.
>
> ---
>
> **Reviewer**: How much of the overall analysis depends on Gaussianity? I suspect the robust gradient descent analysis can be easily extended.
>
> **Authors**: The conditions we require on the (clean) distribution of $a_i$ are: an upper bound on its tail (used in the proof of Lemma 4.2) and bounded first few moments (used in the proof of Lemma 5.1). Therefore, our results directly extend to isotropic sub-Gaussian distributions and other distributions with these properties.

---

> > ### Comment · Reviewer_WB9r · 2025-08-04
> >
> > Thank you very much for your response. I encourage you to consider alternative contamination models and/or extend your impossibility result, if possible. I will maintain my original score.

---

### Official Review · Reviewer_9bGY · 2025-07-03

**Clarity:** 2
**Significance:** 2
**Originality:** 3
**Rating:** 2
**Confidence:** 3

**Summary:**

This work tackles the challenge of recovering real-valued signals in the phase retrieval problem under adversarial corruption in possibly both sampling vectors and intensity measurements. The proposed method operates in two stages:
1. It first uses a convex optimization procedure to obtain a corruption-resilient initialization.
2. Then refines the estimate through gradient descent combined with a robust mean estimation technique.

A key result of the analysis is that the convergence rate is independent of the corruption level.

**Questions:**

Please see previous sections.

**Ethical Concerns:**

["NO or VERY MINOR ethics concerns only"]

**Final Justification:**

After reading the rebuttal and subsequent discussions, I have decided to maintain my score.

Unresolved concerns:
I remain unconvinced regarding the dependence between $\epsilon$ and $n$.

**Limitations:**

yes

**Paper Formatting Concerns:**

-

**Quality:**

2

**Strengths And Weaknesses:**

The authors address an interesting problem that is relevant to both the robust statistics and signal processing communities. However, the paper is quite dense and can be difficult to follow at times. Given the theoretical nature of the contributions, it is especially important to maintain clearer bookkeeping of the key variables involved in the theoretical guarantees.

The central claim of the paper is that the results are independent of the corruption level $\epsilon$. For instance, in lines 93–99, the authors assert that $\epsilon_0$ is an absolute constant, independent of both $n$ and $d$. However, this assertion does not appear to be fully justified. A closer examination of the statement of Lemma 2.3 suggests otherwise.

1. It is evident that the statement of Lemma 2.3 does not hold with high probability when $\epsilon$ is very small, for example, when $\epsilon = 0$. This issue can be addressed by assuming $\epsilon \leq \epsilon_0$, allowing one to replace $\epsilon$ with $\epsilon_0$. However, this introduces a practical challenge, as the value of $\epsilon_0$ is not explicitly specified.
2. Even setting aside practical considerations, for the statement of Lemma 2.3 to hold with high probability, it is necessary that $\epsilon_0 = \omega(\frac{1}{n})$. This implies that $\epsilon_0$ cannot be any absolute constant independent of $n$. Similar issues arise elsewhere - for example, in Lemma 5.2, where the choice of $m$ must depend on $\epsilon_0$ for the Lemma statement to hold.

The presentation of the results would benefit from explicitly retaining all occurrences of $\epsilon$ (or $\epsilon_0$) in the theoretical statements. Doing so would provide better context and help clarify the dependence of the guarantees on the corruption level.

---

> ### Author Rebuttal · Authors · 2025-07-31
>
> We thank the reviewer for their time spent reading and providing feedback on our work.
>
> ---
>
> The reviewer’s main concern is that “the central claim is that the results of this paper are independent of $\epsilon$” is incorrect. We respectfully disagree. Our results indeed hold as stated, and a careful reading of our proofs confirms this.
>
> Below, we address the reviewer's specific doubts in detail.
>
> ---
>
> **Reviewer**: “It is evident Lemma 2.3 does not hold with high probability when $\epsilon$ is very small”
>
> **Authors**: This is true. However, in our algorithm, we only need to robustly estimate the gradient **up to constant precision** (e.g., $c = 4$ in Lemma 5.3), so we never invoke Lemma 2.3 with very small $\epsilon$.
>
> ---
>
> **Reviewer**: For the statement of Lemma 2.3 to hold with high probability , it is necessary $\epsilon_0 = w(1/n)$. This implies that $\epsilon_0$ cannot be any absolute constant independent of $n$.
>
> **Authors**: An absolute constant is $\omega(1/n)$.
>
> ---
>
> **Reviewer**:  Similar issues arise elsewhere - for example, in Lemma 5.2, where the choice of   m must depend on  epsilon_0  for the Lemma 5.2 statement to hold.
>
> **Authors**: There is nothing wrong with $m$ depending on $\epsilon_0$. We only need a constant approximation when estimating the gradient (e.g., $c = 4$ in Lemma 5.3). This leads to $\mathrm{poly}(1/c)$ dependence in both sample complexity and runtime, but $\mathrm{poly}(1/c) = \Theta(1)$ when $c = 4$.
>
> ---
>
> Our paper provides detailed mathematical proofs for all theorems and lemmas. If there are concerns about correctness, we would appreciate it if the reviewer could point out specific incorrect steps.

---

> ### Author Response · Authors · 2025-08-04
>
> In Lemma 5.2, the only place where Lemma 2.3 is actually used, we invoke Lemma 2.3 with $\epsilon = O(c^2) = O(1)$. Therefore, the reviewer's concern about very small $\epsilon$ (e.g., $\epsilon \leq 1/n \log(1/(1 - \tau))$) is irrelevant to our analysis, as we never use Lemma 2.3 in that regime.
>
> In other parts of the paper (e.g., Lemmas 3.1 and 3.2), we only need $\epsilon$ to be at most a universal constant, which is independent of $n$. We kindly ask the reviewer to point out a specific step in our proofs where $\epsilon$ must be at least some function of $n$. To the best of our knowledge, no such dependence is required.

---

### Comment · Area_Chair_38nN · 2025-08-04
**Please carefully read the rebuttal and start the discussion**

Dear Reviewers and Authors,

Thank you all for your efforts so far. As the author–reviewer discussion period will conclude on **August 6**, please start the discussion as soon as possible.


**For Reviewers:**
Please read the authors’ responses and, if necessary, continue the discussion with them.

* If your concerns have been addressed, consider updating your review and score accordingly.

* If some concerns remain, or if you share concerns raised by other reviewers, clearly state these in your review and consider adjusting your review (positively or negatively).

* If you feel that your concerns have not been addressed, you may also choose to keep your review as is.

* I will follow up with you again during the reviewer–AC discussion period (August 7–13) to finalize the reviews and scores.


**For Authors:**
If you have not already done so, please respond to all questions raised by the reviewers. Keep your responses factual, concise, and ensure that every point raised is addressed.

Best regards,

The AC

---

### Decision · Program_Chairs · 2025-09-17

**Decision:**

Reject

**Comment:**

This paper proposes a nearly linear-time algorithm for robust phase recovery with a novel combination of spectral initialization and robust gradient descent, supported by theoretical analysis. The problem is important and the approach is conceptually simple and potentially generalizable, but the work currently falls short of NeurIPS standards due to presentation clarity, vague or incomplete proofs, and a lack of numerical experiments of comparison to substantiate the theoretical claims. Without stronger validation and clearer exposition, the contribution remains promising but not yet ready for publication.